# Hexokinase 2-driven glycolysis in pericytes activates their contractility leading to tumor blood vessel abnormalities

Ya-Ming Meng[1,2,3,7], Xue Jiang[1,3,7], Xinbao Zhao[1,4,7], Qiong Meng[1,3], Sangqing Wu[1,3], Yitian Chen[1,3], Xiangzhan Kong[1,3], Xiaoyi Qiu[1,3], Liangping Su[1,3], Cheng Huang[1,3], Minghui Wang[5], Chao Liu[6] & Ping-Pui Wong [1,3✉]

Defective pericyte-endothelial cell interaction in tumors leads to a chaotic, poorly organized and dysfunctional vasculature. However, the underlying mechanism behind this is poorly studied. Herein, we develop a method that combines magnetic beads and flow cytometry cell sorting to isolate pericytes from tumors and normal adjacent tissues from patients with non-small cell lung cancer (NSCLC) and hepatocellular carcinoma (HCC). Pericytes from tumors show defective blood vessel supporting functions when comparing to those obtained from normal tissues. Mechanistically, combined proteomics and metabolic flux analysis reveals elevated hexokinase 2(HK2)-driven glycolysis in tumor pericytes, which up-regulates their ROCK2-MLC2 mediated contractility leading to impaired blood vessel supporting function. Clinically, high percentage of HK2 positive pericytes in blood vessels correlates with poor patient overall survival in NSCLC and HCC. Administration of a HK2 inhibitor induces pericyte-MLC2 driven tumor vasculature remodeling leading to enhanced drug delivery and efficacy against tumor growth. Overall, these data suggest that glycolysis in tumor pericytes regulates their blood vessel supporting role.

[1] Guangdong Provincial Key Laboratory of Malignant Tumor Epigenetics and Gene Regulation, Guangdong-Hong Kong Joint Laboratory for RNA medicine, Sun Yat-sen Memorial Hospital, Sun Yat-sen University, 510120 Guangzhou, China. [2] Department of Obstetrics and Gynecology, Center for Reproductive Medicine, Key Laboratory for Major Obstetric Diseases of Guangdong Province, The Third Affiliated Hospital of Guangzhou Medical University, 510150 Guangzhou, China. [3] Medical Research Center, Sun Yat-sen Memorial Hospital, Sun Yat-sen University, 510120 Guangzhou, China. [4] Department of Ultrasound, Sun Yat-sen Memorial Hospital, Sun Yat-sen University, 510120 Guangzhou, China. [5] Department of Thoracic surgery, Sun Yat-sen Memorial Hospital, Sun Yat-sen University, 510120 Guangzhou, China. [6] Department of Biliary-Pancreatic Surgery, Sun Yat-sen Memorial Hospital, Sun Yat-sen University, 510120 Guangzhou, China. [7] These authors contributed equally: Ya-Ming Meng, Xue Jiang, Xinbao Zhao. ✉email: huangbp3@mail.sysu.edu.cn

Pericytes have an important regulatory role in stabilizing blood vessel wall, controlling vessel dilation, and regulating vascular perfusion[1–3]. Unlike other host cells, pericyte expresses specific surface markers/receptors such as platelet derived growth factor receptor β (PDGFRβ) and CD146 (also known as the melanoma cell adhesion molecule (MCAM)), and respond to specific growth factors including PDGF, transforming growth factors and angiopoietins[2,4]. During angiogenesis, endothelial cells secrete PDGF to recruit pericytes to newly formed vessels via activation of the PDGFRβ signaling[1,5]. Unlike PDGFRβ, CD146 was originally identified as an endothelial cell marker during angiogenesis[6], which has been recently shown to highly express in pericytes too, and important for regulating pericyte-endothelial cell interaction during blood–brain barrier development[4]. Therefore, these two surface proteins are routinely used as positive markers for pericyte identification.

Just as tumor endothelial cells differ from quiescent endothelial cells in normal tissues, tumor-derived pericytes appear to be abnormal or dysfunctional when compared with pericytes in normal tissues[7,8]. It is unclear whether cancer cells instruct the pericyte fate changes to facilitate tumor growth. Indeed, defective pericyte-endothelial cell interaction is observed in many cancers, which is characterized by compromised blood vessel integrity[9,10]. For instance, the loss or detachment of pericytes from tumor blood vessels is frequently observed in some cancers and is believed to promote cancer metastasis[11,12]. On the other hand, pericytes have been shown to protect endothelial cells from anti-angiogenic drug (i.e. bevacizumab), and therefore combination treatment with PDGFRβ tyrosine kinase inhibitor (i.e. eliminating PDGFRβ positive pericytes) and VEGF inhibition more efficiently block tumor angiogenesis than anti-VEGF therapy alone in several animal models[13,14]. However, the combination therapy has failed to provide any therapeutic benefit in the clinical trial of renal carcinoma patients and even shows additive toxicity in the patients[14,15]. Therefore, it is a clinically unmet need to explore the molecular mechanism of pericyte-endothelial cell interaction in preclinical setting. Due to the lack of a cellular model for studying human pericyte biology, the exact cause of abnormal pericyte behavior in tumors is unknown. Previous studies show that the regulation of metabolic pathways determines the cell state and fate decisions in endothelial cells[16]. For instance, Inhibition of the glycolytic activator PFKFB3 in endothelial cells can improve pericyte coverage and blood vessel function, which therefore enhances the delivery and efficacy of chemotherapeutics drugs in preclinical animal models[17]. However, the role of metabolic rewiring in human pericyte biology still remains largely unknown.

In this study, we establish the role of pericyte-hexokinase 2 (HK2) driven glycolysis in remodeling tumor vasculature. By developing a method for isolation and culture of pericytes from normal adjacent tissues (NPC) and tumors (TPC) derived from NSCLC (non-small-cell lung cancer) and HCC (hepatocellular carcinoma) patients respectively, we showed that TPC had abnormal blood vessel supporting functions as compared with NPC in two different cancers. Combined proteomics and metabolic flux analysis revealed an increased HK2-driven glycolysis in TPC as compared with NPC, which prohibited its blood vessel supporting role via activation of the ROCK2-MLC2 mediated contractility. Importantly, depletion of HK2/ROCK2 restored the blood vessel supporting function of TPC, while treatment of HK2 inhibitor induces MLC2-driven tumor vasculature remodeling to enhance chemotherapy delivery and efficacy against tumor growth. The role of pericyte-HK2 expression in this process was further illustrated by the fact that administration of chemotherapy reduced tumor growth in mice that were co-injected with tumor cells and HK2-depleted TPC as compared with mice that were co-injected with tumor cells and scramble transfected TPC.

Overall, our work develops a cellular model for studying pericyte biology and uncovers that pericyte-HK2-driven glycolysis induces tumor blood vessel abnormality by activating ROCK2-MLC2 mediated contractility.

## Result

**Development of the microbead activated vascular cell sorting enrichment method coupled with FACS.** Although previous studies have managed to isolate pericytes from mouse brain tissues[18,19], there is no effective protocol for isolation and culture of human pericytes from paired normal adjacent tissues and tumors. Due to the fact that both lung and liver tissues contain a rich population of vascular cells, they were therefore chosen as a source of human pericyte isolation in this study (Supplementary fig. 1). Since anti-human CD146 antibody conjugated with FITC and anti-FITC magnetic microbeads are commercially available, we proposed to employ a CD146+-FITC-microbead activated cell sorting (MACS) method to enrich the vascular cell population from tissue samples first, and then used positive and negative selection markers to isolate pericyte population via fluorescence activated cell sorting (FACS) (Supplementary fig. 1). In order to carry out our pericyte purification method, we first examined whether CD146 was expressed on vascular endothelial cells and pericytes in normal adjacent tissue and tumors derived from NSCLC and HCC patients. Based on the fact that CD34 and α-SMA/PDGFRβ is known as a vascular endothelial cell and pericyte marker respectively[7,20], they were used for defining the CD146 staining in this study. We then performed triple immunofluorescent staining of these markers with paired normal adjacent tissues and tumors derived from NSCLC and HCC patients respectively, showing that CD146 was expressed in both CD34 positive (+ve) vascular endothelial cells and αSMA/PDGFRβ+ve pericytes within normal adjacent tissues and tumors derived from NSCLC and HCC patients, while ~65–72% percentage of blood vessels were CD146 and α-SMA double positive in both normal adjacent tissues and tumors derived from NSCLC and HCC patients respectively (Fig. 1a–f and Supplementary fig. 2a, b). No significant difference in the percentage of CD146 and α-SMA double positive blood vessels was observed between normal adjacent tissues and tumors (Fig. 1c, f). In addition, we showed that α-SMA was also expressed in PDGFRβ+ve pericytes around CD34 positive blood vessels (Supplementary fig. 2c, d), which was consistent with the previous finding[1]. Therefore, our results suggested that CD146+-FITC-microbead activated cell sorting method could be used to enrich the vascular cell population from tissue samples derived from NSCLC and HCC patients.

To isolate pericytes from normal adjacent tissues and tumors, the dead cells and debris were first removed from mechanically minced and enzymatically digested tissues. The single cell suspension was first labeled with FITC-conjugated CD146 antibody and then enriched by using anti-FITC magnetic microbeads. The enriched vascular cell population was subsequently labeled with antibodies specific for endothelial cells (CD31), immune cells (CD45), and pericyte markers (PDGFRβ). The labeled cells were then collected by using FACS (Supplementary fig. 1). FACS plots showed the gating strategy for the separation of endothelial cells (EC) and pericytes (PC) from normal adjacent tissues (NEC and NPC) and tumor tissues (TEC and TPC) derived from NSCLC and HCC (Fig. 1g–j and Supplementary fig. 2e–h). Importantly, our MACS enrichment method successfully improved the efficacy of isolating human pericytes from paired normal adjacent

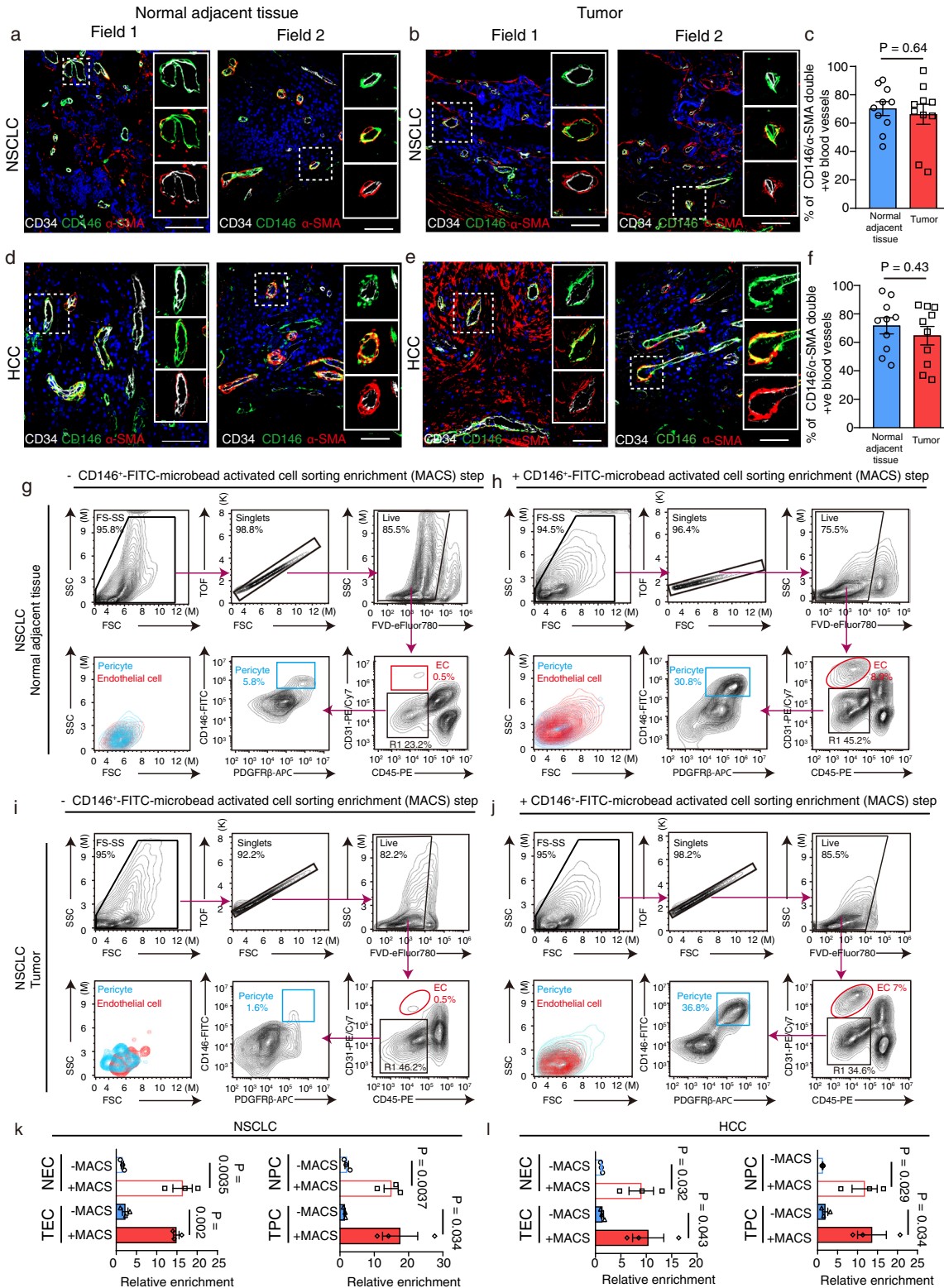

tissues and tumors derived from NSCLC and HCC patients almost by 7–13 folds as compared to FACS method alone (Fig. 1k, l). Compared with FACS method alone, our protocol enhanced the successful chance of isolating enough pericytes for subsequent culturing and characterization and functional studies (Supplementary fig. 1). Overall, this method enabled us to obtain pure pericytes in culture with high reproducibility, mainly due to MACS enrichment method and temperature/

time-controlled incubation with optimized amount of collagenase and DNase I.

**Characterization of the isolated pericytes from normal adjacent tissues and tumors derived from NSCLC and HCC patients.** To confirm the purity of our isolated pericytes derived from normal adjacent tissues and tumors, we performed flow

**Fig. 1 Purification of pericytes from normal adjacent tissues and tumors derived from NSCLC and HCC patients using CD146$^{+}$-FITC-MACS enrichment method. a–f** Representative triple immunostaining images of CD34 (white), CD146 (green), and α-SMA (red) from 2 different microscopic fields in normal adjacent tissues and tumors derived from NSCLC and HCC patients are given. Bar charts represent the percentage of CD146 and α-SMA-double positive (+ve) blood vessels per field in each group (n = 10 NSCLC/HCC patients). Statistical tests were two-sided. **g–j** Representative FACS plots showing the gating strategy for the purification of pericytes from normal adjacent tissues and tumors derived from NSCLC patients with (+) or without (−) CD146$^{+}$-FITC-microbead cell soring enrichment (MACS) method. Debris, doublets (TOF), and dead cells (FVD) were excluded, then CD45$^{-}$CD31$^{-}$CD146$^{+}$PDGFRβ$^{+}$ pericytes were collected. Percentage refers to the proportion of cells in the previous parent gate. **k, l** Bar charts show the relative fold change in percentage of pericytes and endothelial cells isolated by the MACS enrichment coupled with FACS protocol compared to that isolated by FACS alone. Statistical tests were two-sided. Percentages of pericyte refer to cell population in the R1 gate. NSCLC non-small cell lung cancer, HCC hepatocellular carcinomas, FSC forward-scatter; SSC side-scatter, TOF time of fly, FVD fixable viability dye, EC endothelial cells, PC pericytes, NEC normal adjacent tissue derived endothelial cells, TEC tumor-derived endothelial cells, NPC normal adjacent tissue derived pericytes, TPC tumor-derived pericytes. FACS plots/bar charts are representative of three individual patient data. Means ± SEM are given. **c**, **f**, **k**, **l** Student's *t* test. Scale bars in **a**, **b**, **d**, **e** represent 50 μm, **a**, **b**, **d**, **e** (magnified pictures) represent 25 μm.

cytometry and immunostaining experiments with our isolated pericytes, indicating that the TPC and NPC derived from NSCLC or HCC patients expressed PDGFRβ and CD146 as well as newly identified pericyte markers desmin and CD13, but did not express endothelial cell marker CD31 and CD34, immune cell marker CD45, or fibroblast marker FAP (fibroblast associated protein) (Fig. 2a, b and Supplementary fig. 3a, b). Further RT-PCR analysis confirmed that these pericytes either expressed relatively low or undetectable levels of endothelial cell markers *CD31* and *CD34*, immune cell marker *CD45*, smooth muscle cell markers *CNN1* and *MYOCD*, and pericyte/fibroblast marker *ACTA2*, fibroblast markers *FAP* and *PDGFRA* (platelet derived growth factor α) as compared with human umbilical vein endothelial cells (HUVEC), peripheral blood mononuclear cell (PBMC), and human fibroblasts respectively (Fig. 2c and Supplementary fig. 3c). In contrast, both NSCLC/HCC-derived NPC and TPC strongly expressed pericyte markers *PDGFRB* and *CD146* as compared to fibroblasts and HUVEC respectively (Fig. 2c and Supplementary fig. 3c). In addition, no significant difference in the expression level of pericyte markers between either NPC and TPC or TPC at passage 3 and 12 was observed (Fig. 2a–c and Supplementary fig. 3a–e), suggesting that our culturing method could maintain the pericyte identity of the isolated cells over passages.

Next, we examined their cell morphology, proliferation, doubling time, passage number, senescence, viability, and migration ability. Phase contrast microscope analysis revealed that both NPC and TPC had the morphological characters of pericytes as observed previously in tissue section, which possessed a cell body with a prominent nucleus and a small content of cytoplasm with several long processes (Fig. 2b and Supplementary fig. 3b). Functional studies revealed that the NPC and TPC derived from NSCLC or HCC patients had similar doubling times and could be cultured up to 12 passages with gradual senescence observed during 8–12 passages (Fig. 2d, e), while there was no significant difference in proliferation (Supplementary fig. 3f, g), cell viability (Supplementary fig. 3h), and migration (Fig. 2f, g) between NPC and TPC. To determine whether NPC and TPC showed any difference in endothelial cell dependent recruitment, we plated NPC or TPC in the upper compartment of a transwell, while HUVEC cells were placed into the bottom compartment, indicating that NPC and TPC displayed no difference in response to HUVEC dependent recruitment (Supplementary fig. 3i). Taken together, our work developed a cellular model to study the functional differences between NPC and TPC.

**Tumor-derived pericyte possesses defective blood vessel supporting role.** Pericytes have an important regulatory role in stabilizing blood vessel wall by regulating endothelial cell properties[1,9]. To determine the role of NPC and TPC in

endothelial cell proliferation and migration, we exposed HUVEC cells to conditioned medium from NPC/TPC derived from NSCLC or HCC patients, indicating that the conditioned media from either NPC or TPC increased the proliferation of HUVEC (Fig. 3a), while neither of them showed any effect on HUVEC migration in a wound healing assay (Fig. 3b). To examine the functional role of NPC and TPC in blood vessel formation, we fluorescently labeled NPC and TPC as well as fibroblasts and smooth muscle cells (SMC), and then co-cultured HUVEC with either of them on matrigel. After co-culturing, we evaluated their effect on total tube length, branch points, and numbers (Fig. 3c–f and Supplementary fig. 4a–d). Although NPC, TPC, fibroblasts, and smooth muscle cells aligned along tubes when added to HUVECs, they had different effects on tube lengths and branching. Strikingly, co-culturing HUVEC with TPC reduced total tube length and branch points and number as compared to HUVEC co-cultured with either NPC, fibroblasts, or HUVEC alone, while normal fibroblasts had no effect on all of these (Fig. 3c–f and Supplementary fig. 4a–d). Unlike NPC, TPC or normal fibroblast, SMCs pulled tubes apart following alignment and caused a rapid regression of the tubular network within 24 h (Fig. 3c–f and Supplementary fig. 4a–d).

To evaluate the blood vessel supporting role of NPC and TPC in vivo, we performed in vivo matrix plug assays by embedding NPC/TPC/fibroblasts/SMC in matrigel-fibrin matrix with HUVEC, and then implanted them into nude mice subcutaneously. Matrix gel plug assays indicted that co-transplantation of HUVEC and NSCLC/HCC-derived-NPC and-TPC significantly increased the number of blood vessels as compared with HUVEC transplant alone (Fig. 3g, h). Further histological examination showed that co-transplantation of HUVEC with NPC, but not TPC, supported the formation of functional blood vessels (containing red blood cells) with strong basement membrane marker collagen IV expression (Fig. 3g–j), while smaller and irregular shaped blood vessels, reduced collagen IV staining were observed in HUVEC co-transplanted with TPC (Fig. 3g–j). Previous studies have shown that the expression of basement membrane marker collagen IV has been implicated in blood vessel integrity and perfusion[1,7]. As expected, both smooth muscle cells and fibroblasts had a very limited effect on tumor blood vessel formation and collagen IV expression (Fig. 3g, h, j), while SMC reduced the lumen diameter of blood vessels most significantly as compared with TPC and NPC (Fig. 3g, i). Finally, neither NPC, TPC, fibroblast, or SMC showed difference in tube coverage (Fig. 3k).

Since pericytes have been implicated in regulating capillary dilation and contraction[1,14], we therefore assessed the contractile properties of NPC and TPC by performing time-lapse microscopy studies with them after treated with carbachol. After the treatment, NSCLC/HCC-derived TPC strongly contracted and

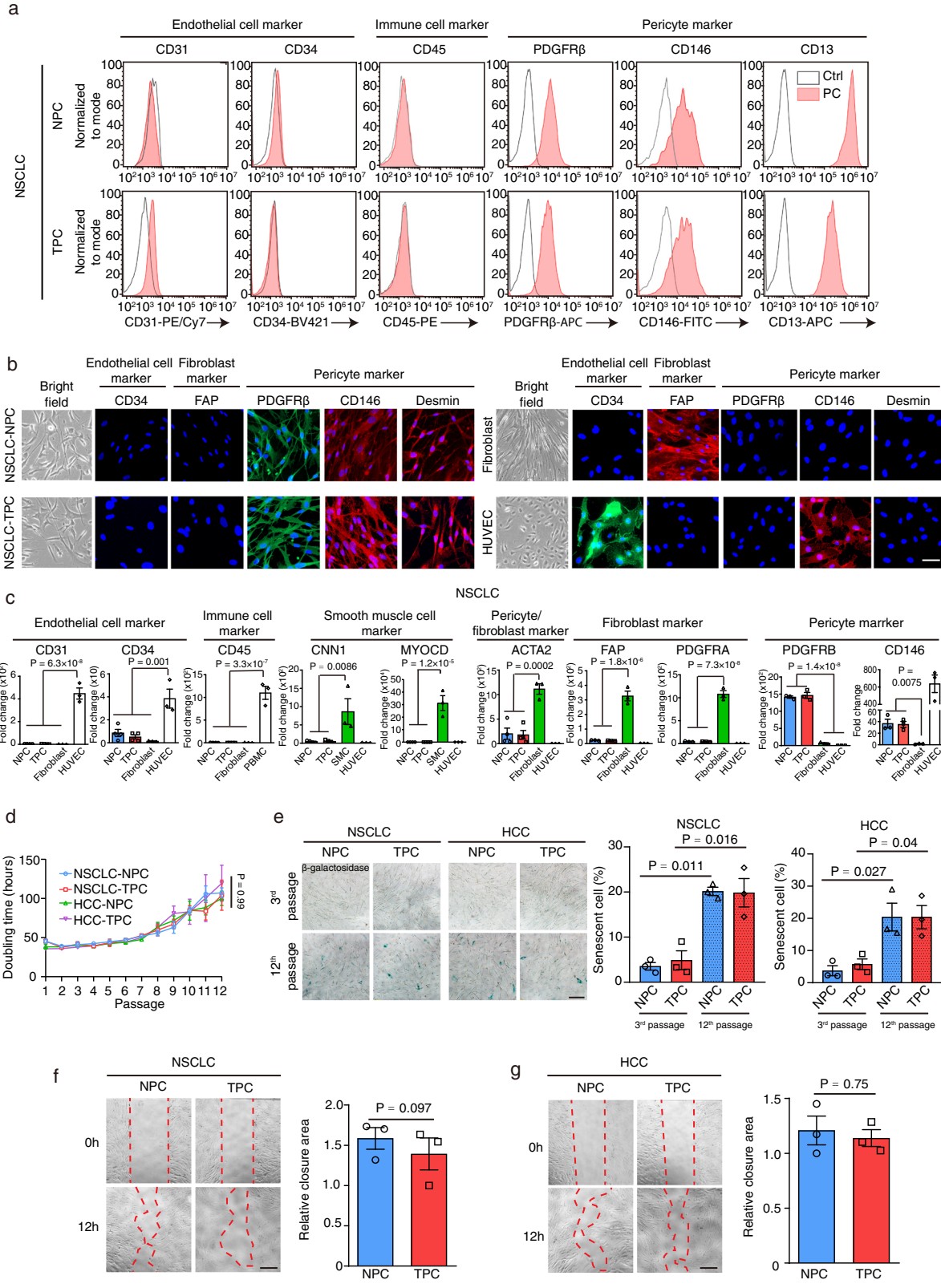

showed up to a 30% change in surface area, while NSCLC/HCC-derived NPC displayed a modest change in their surface area (Supplementary fig. 4e, f). In contrast, no significant change in the surface area of HUVEC/fibroblast was observed post-treatment (Supplementary fig. 4e, f). Together, these results showed the functional difference between NPC and TPC in vitro and in vivo, indicating that TPC had poor BV supporting functions as

compared with NPC, which may be linked to its increased contractility phenotype.

**Combined proteomics and metabolic flux analysis reveals an elevated HK2-driven glycolysis in TPC.** Metabolic reprogramming determines cell proliferation, migration, and contractility[21]. Indeed, endothelial cells undergo PFKFB3 dependent glycolytic

**Fig. 2 Characterization of normal adjacent tissue derived pericytes and tumor-derived pericytes. a** Flow cytometry analysis of pericytes isolated from normal adjacent tissue and tumor-derived from NSCLC patients showed that the purity of pericyte preparation was high. Pericyte preparations displayed good expression of pericyte markers PDGFRβ, CD146, and CD13 and undetected or low level of endothelial cell markers CD31 and CD34 as well as immune cell marker CD45. **b** Immunofluorescent staining analysis of the endothelial cell, fibroblast, pericyte marker expression in NPC, TPC, HUVEC, and fibroblast (*n* = 3 independent samples). **c** RT-PCR analysis of the endothelial cell, fibroblast, smooth muscle cells, pericyte marker expression in NPC, TPC, HUVEC, fibroblasts, peripheral blood mononuclear cell (PBMC), smooth muscle cells (SMC) (for *CD31, CD34, CD45, CNN1, MYOCD,* and *ACTA2* expression analysis: *n* = 4 independent NPC and TPC samples and *n* = 3 independent SMC, PBMC and HUVEC samples; for *FAP, PDGFRA, PDGFRB,* and *CD146* expression analysis: *n* = 3 independent samples for each cell type). **d** Doubling time of NPC and TPC derived from NSCLC/HCC patients (*n* = 3 independent experiments). **e** β-galactosidase staining assay showed the percentage of senescent cells at different passages of NSCLC/HCC-derived NPC and TPC (*n* = 3 independent experiments). Statistical tests were two-sided. **f, g** Wound healing assay showed no difference in the migration ability of NSCLC/HCC-derived NPC and TPC (*n* = 3 independent experiments). Statistical tests were two-sided. Means ± SEM are given. **c** One-way ANOVA. **d** Two-way ANOVA. **e–g** Student's *t* test. Scale bars in **b** represent 50 μm, **e–g** 100 μm.

switch to match their high energy demand for migration and proliferation during angiogenesis. However, the role of metabolic state in determining pericyte's BV supporting function remains unknown. In addition, it is unclear whether NPC and TPC express different glycolytic gene signature. We therefore performed RT-PCR analysis of the NPC and TPC isolated from NSCLC or HCC patients to compare their glycolytic gene pathway expression, indicating that NPC and TPC had distinct difference in the expression of genes associated with glycolysis (Fig. 4a, b). In both NSCLC and HCC, the expression of glycolytic gene, hexokinase 2 (*HK2*) was significantly up-regulated in TPC as compared to NPC (Fig. 4a, b). Although previous studies indicated that inhibition of PFKFB3 in human placenta pericytes prohibited their cell migration and proliferation abilities[17], we did not observe any difference in the *PFKFB3* expression between NPC and TPC (Fig. 4a, b). In addition, proteomics analysis indicated that the expression of HK2, but not PFKFB3 as well as other glycolytic enzymes and glucose transporters, was significantly increased in TPC as compared to NPC (Fig. 4c). Western blot analysis confirmed the up-regulation of HK2 protein expression in NSCLC/HCC-derived TPC as compared with NPC (Fig. 4d). Based on these findings, we next examined the glycolytic capacity of NPC and TPC by performing glucose uptake assay, lactate assay, Seahorse analysis as well as metabolic flux analysis. As expected, NPC and TPC showed no difference in the glucose uptake, while the lactate production was significantly increased in TPC as compared with NPC (Fig. 4e, f). Further seahorse assay indicated an increase in extracellular acidification rate (ECAR) but decreased oxygen consumption rate (OCR) in TPC as compared that to NPC (Fig. 4g, h and Supplementary fig. 5a, b), implying that TPC relied on glycolysis for their energy production. To obtain the mechanistic underpinnings of the deregulated TPC metabolism, we carried out $^{13}$C-labeled metabolic flux analysis using U-$^{13}$C$_6$ glucose to estimate metabolic fluxes in TPC and NPC derived from NSCLC or HCC patients. Metabolic tracking revealed the relative abundance of glycolysis metabolite M3 lactate was increased in TPC as compared that to NPC, while no significant difference in the relative abundance of TCA (tricarboxylic acid) cycle metabolites directly derived from U-$^{13}$C$_6$ glucose, such as M2 citrate, M2 ketoglutarate, M2 succinate, M2 fumarate, and M2 malate, was observed between NPC and TPC (Fig. 4i and Supplementary fig. 5c).

To prove the clinical significance of our observation, we performed an immunofluorescence study with normal adjacent tissues and tumors derived from NSCLC and HCC patients, indicating that the percentage of pericyte-HK2 positive blood vessels was significantly up-regulated in tumors as compared to normal adjacent tissues (Fig. 4j and Supplementary fig. 5d). We next performed triple immunostaining of PDGFRβ, CD34, and α-SMA with our own NSCLC and HCC patient cohorts, indicating that high percentage of pericyte-HK2 positive blood vessels

(i.e. when more than the mean number of tumor blood vessels are pericyte-HK2 positive blood vessels across individual patient samples in our NSCLC and HCC cohorts respectively) was associated with poor overall survival in NSCLC and HCC patients respectively (Fig. 4k and Supplementary fig. 5e). Collectively, our findings establish a major metabolic reprogramming of TPC, evidently through up-regulated HK2-driven glycolysis in TPC as compared to NPC.

**Pericyte-HK2-driven glycolysis inhibits the blood vessel supporting function of pericyte via ROCK2-MLC2 mediated contractility.** Next, the function relevance of glycolytic switch in TPC was examined. We pre-treated TPC with placebo or HK2 inhibitor for 15 min and then co-cultured them with HUVEC cells in tube formation assays. In both NSCLC and HCC, we showed that pre-treatment with HK2 inhibitor enhanced the total tube length, branch points, and number in HUVEC co-cultured with TPC, when compared with HUVEC co-cultured with placebo pre-treated TPC (Fig. 5a–d). To test its effect in vivo, we embedded HUVEC with TPC pre-treated with placebo or HK2 inhibitor 3-bromopyruvate (3-BP) in matrigel-fibrin matrix, which were then implanted into nude mice and given either placebo or HK2 inhibitor treatment before harvesting. Strikingly, treatment with HK2 inhibitor did not affect the capacity of TPC for promoting blood vessel formation, while it enhanced blood vessel diameter and collagen IV expression in HUVEC co-transplanted with TPC as compared to HUVEC co-transplanted with placebo pre-treated TPC (Fig. 5e–h). No significant difference in percentage of coverage between these two groups (Fig. 5i). Importantly, treatment with HK2 inhibitor did not affect blood vessel density, blood vessel diameter, and collagen IV expression in HUVEC transplant alone as compared with placebo treated group (Fig. 5e–h). Further metabolic study showed that pre-treatment with HK2 inhibitor had no effect on glucose uptake in TPC as compared with placebo treated TPC (Supplementary fig. 6a), while it significantly inhibited lactate production in TPC (Supplementary fig. 6b). In addition, pre-treatment with HK2 inhibitor significantly reduced carbachol-induced cell contraction in TPC as compared to placebo pre-treated TPC (Supplementary fig. 6c). These results demonstrated that the elevated HK2 expression in TPC contributed to its defect in cell contractility and blood vessel supporting function, and that was determined by glycolysis.

Consistent with these findings, Rho-associated coiled-coil containing protein kinase (ROCK) signaling pathway has been implicated in regulating pericyte shape and contractility[22,23], while these processes require energy supply mainly from glycolysis[24]. Non-muscle myosin II is a hexameric actin-binding protein that is formed by two heavy chains and two regulatory light chains, which has an important role in cell contractility[25]. Interestingly, ROCK can directly phosphorylate MLC2 to increase

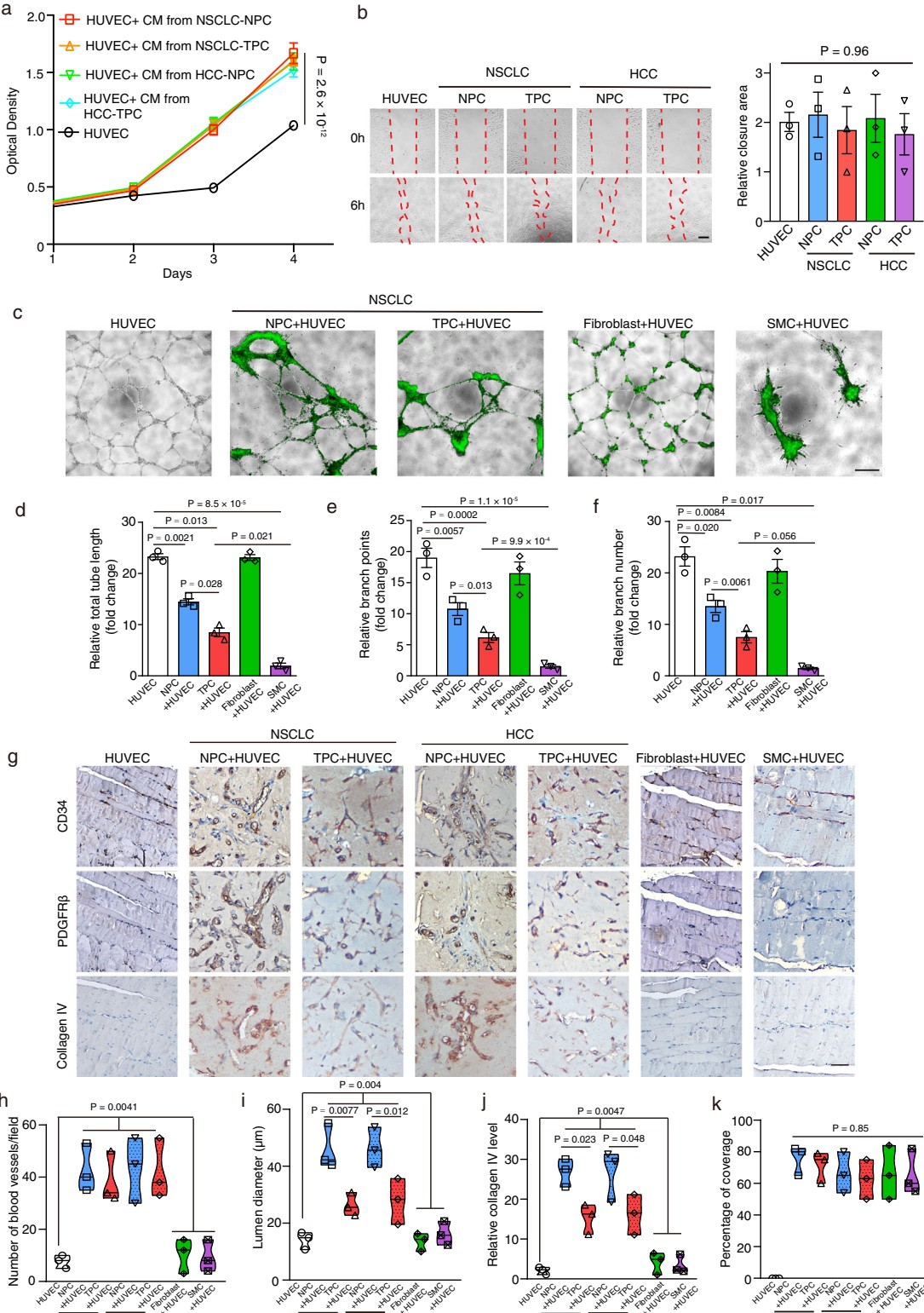

**Fig. 3 Tumor-derived pericytes display defective blood vessel supporting functions in vitro and in vivo. a** CCK8 proliferation assay showed that exposure with conditioned medium from NSCLC/HCC-derived NPC or TPC significantly enhanced the proliferation of HUVEC as compared to untreated HUVEC ($n = 5$ experimental repeats). **b** Wound healing assay of the HUVEC after exposed with condition medium from NSCLC/HCC-derived NPC or TPC. Representative pictures of wound healing assay in each group are given ($n = 3$ independent experiments). **c** HUVEC were either cultured alone or co-cultured with CFSE (green fluorescent dye) labeled NSCLC-derived-NPC/-TPC, fibroblast or smooth muscle cells (SMC) in a tube formation assay. **d–f** Bar charts show the relative total tube length, branch points and number in each group ($n = 3$ independent experiments). **g** Representative images of the serial sections of matrix plugs after immunostained with CD34, PDGFRβ, or collagen IV antibody in each group are given. **h–k** Violin plots show the quantification of the total number of blood vessels, lumen diameter, collagen IV level and percentage of coverage in each group ($n = 3$ mice per group). Results are given as means ± SEM. **a** Two-way ANOVA. **b**, **d–f**, **h–k** One-way ANOVA. Scale bars in **b** represents 100 μm, **c** 200 μm, **g** 50 μm.

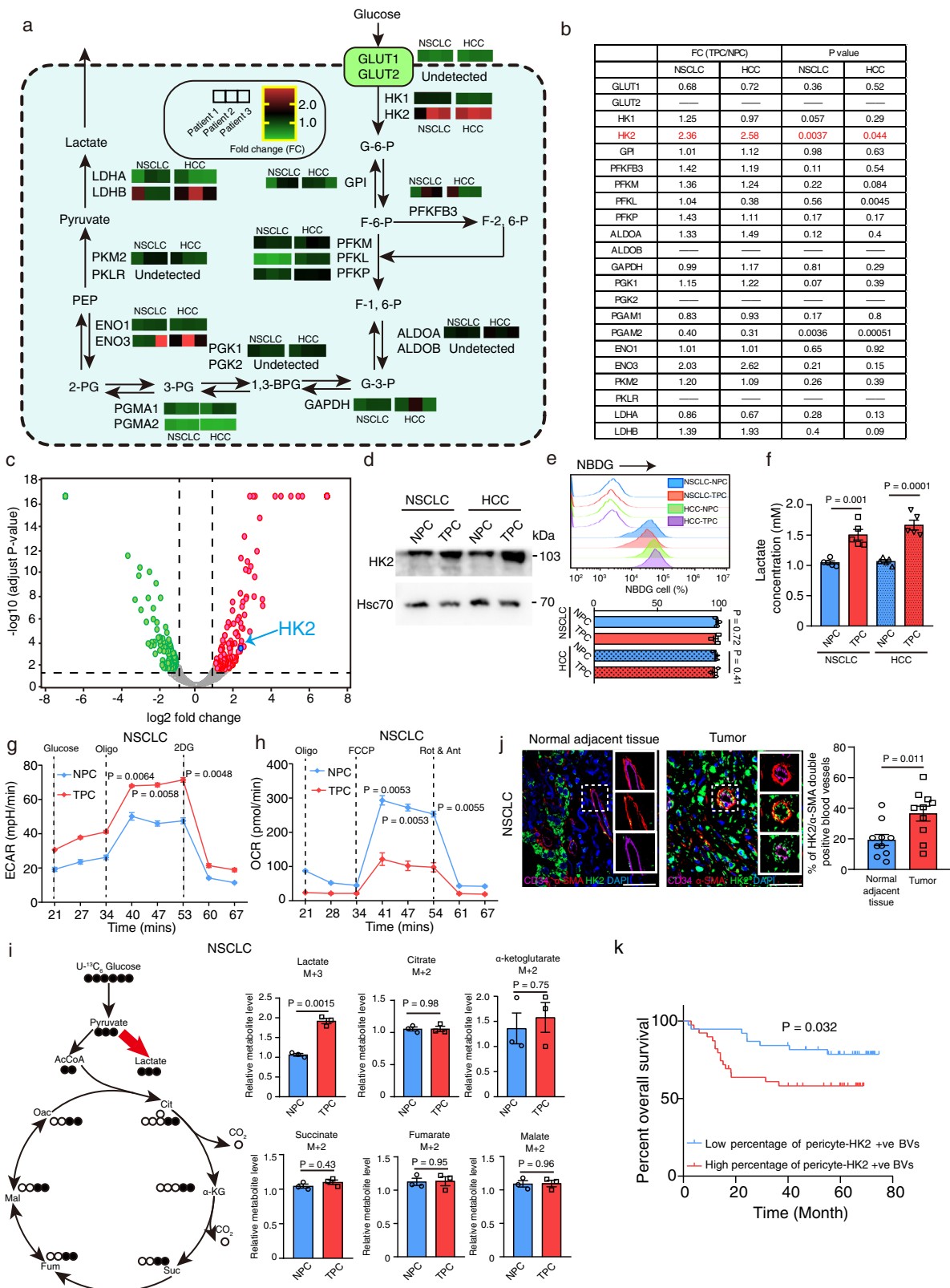

the activity of myosin II during cell contraction[26]. As expected, western blot analysis indicated that the expression of ROCK2 and phosphorylation of ROCK2 downstream effector MLC2 (Myosin-light-chain 2) was increased in TPC as compared to that in NPC (Fig. 5j). Importantly, administration of HK2 inhibitor significantly down-regulated the expression of ROCK2 and p-MLC2 in TPC (Fig. 5k). To determine the role of ROCK2 in regulating

TPC's BV supporting function, in vitro tube formation assay was performed by co-culturing HUVEC with NSCLC/HCC-derived TPC pre-treated with either placebo or ROCK inhibitor GSK 429286A in matrigel, indicating that pre-treatment with ROCK inhibitor significantly enhanced total tube length, branch point, and number (Fig. 5l–o). Furthermore, pre-treatment with ROCK inhibitor also prohibited carbachol-induced cell contraction in

**Fig. 4 Combined proteomics and metabolic flux analysis reveals an elevated HK2-driven glycolysis in TPC as compared with NPC. a** Metabolic network shows the metabolic reactions involved in glycolysis (based on Kyoto Encyclopedia of Genes and Genomes pathway, KEGG), together with main metabolites and 22 selected glycolytic genes. Heatmap analysis of the 22 selected glycolytic gene expression in NSCLC/HCC-derived NPC and TPC ($n = 3$ individual patients). **b** Table shows the average fold change and statistical analysis of 22 glycolytic gene expression between NPC and TPC. **c** Proteomics analysis showed that hexokinase 2 (HK2) was significantly up-regulated in NSCLC-derived TPC as compared with NPC. The x-axis of the volcano plot shows the $\log_2$ fold change of each identified proteins and the y-axis represents the $-\log_{10}$ of P-values. Red dots represent significant up-regulated proteins and green dots for significant down-regulated proteins ($p < 0.05$, fold change more than 2 ($\log_2 = 1$)). The horizontal dot line indicates raw P-value = 0.05. The dots having a fold-change less than 2 ($\log_2 = 1$) and/or P-value <0.05 are shown in gray. Statistical tests were two-sided. **d** Western blotting analysis of the HK2 expression in NSCLC/HCC-derived NPC and TPC. Hsc70 was used as loading control ($n = 3$ independent experiments). **e** The glucose uptake of NSCLC-derived NPC and TPC was measured by using glucose uptake cell-based assay kit ($n = 3$ experimental repeats). Statistical tests were two-sided. **f** The lactate production of NSCLC-derived-NPC and -TPC was measured by glycolysis cell-based assay kit ($n = 5$ experimental repeats). Statistical tests were two-sided. **g, h** The extracellular acidification rate (ECAR) and oxygen consumption rate (OCR) of NSCLC-derived-NPC and -TPC were measured by a Seahorse flux analyser ($n = 5$ independent experiments). Dot lines indicate the time and period of glucose, oligomycin (Oligo), 2-Deoxy-D-glucose (2-DG), carbonyl cyanide-p-trifluoromethoxyphenylhydrazone (FCCP), rotenone (ROT), or antimycin (Ant) being added during seahorse experiments. **i** Schematic diagram represents intracellular fluxes in NSCLC-derived TPC relative to NPC quantified using U-$^{13}C_6$ glucose tracer. Open circle represents $^{12}C$ and filled black circles for $^{13}C$ atoms. Bar charts represent the relative abundance of metabolites between NPC and TPC ($n = 3$ experimental repeats). Red arrow represents increased fluxes in TPC as compared with NPC. Black arrows represent no significant change in fluxes between NPC and TPC. Statistical tests were two-sided. **j** Representative images of the triple immunostaining of CD34 (magenta), HK2 (green), and α–SMA (red) in normal adjacent tissues and tumors derived from NSCLC patients. Bar charts represent the percentage of HK2/α–SMA double positive blood vessels per field in each group ($n = 10$ NSCLC patients). Statistical tests were two-sided. **k** Kaplan–Meier survival study showed that NSCLC patients with high percentage of pericyte-HK2 + ve blood vessels were correlated with poor overall survival ($n = 77$ NSCLC patients). Statistical tests were two-sided. Results are given as means ± SEM. **b**, **c**, **e–j** Student's t test. **k** Log-rank (Mantel-Cox) test. Scale bars in **j** represents 50 μm.

TPC (Supplementary fig. 6c). Western blot analysis indicated that treatment with ROCK inhibitor down-regulated the phosphorylation of MLC2 in TPC as compared with placebo-treated TPC (Supplementary fig. 6d). Clinically, the pericyte expression of ROCK2 was up-regulated in NSCLC/HCC-derived tumors as compared with normal adjacent tissues (Supplementary fig. 6e), while high ratio of ROCK2 to pericyte marker expression correlated with poor survival in NSCLC and HCC patients respectively (Supplementary fig. 6f, g).

Since human ROCKs consist of two isoforms, ROCK1 and ROCK2[26], we further examine whether HK2 regulates TPC's blood vessel supporting role via ROCK1 or ROCK2. We first stably knock-downed HK2, ROCK1, or ROCK2 expression in NSCLC/HCC-derived TPC, and also overexpressed ROCK2 in HK2-depleted TPC, which was confirmed by the RT-PCR and western blot analysis (Supplementary fig. 6h–k). Interestingly, depletion of HK2 down-regulated ROCK2, but not ROCK1 expression, in NSCLC/HCC-derived TPC (Supplementary fig. 6h). Further tube formation assays showed that stable knock-down of pericyte-HK2 expression or depleting pericyte-ROCK2 by shRNA/ROCK inhibitor HA-1077 (i.e. a more potent ROCK2 inhibitor[27]) increased the total tube length, branch points, and number in HUVEC co-cultured with TPC as compared with HUVEC co-cultured with scramble transfected TPC, while silencing ROCK1 had no significant effect on tube formation (Supplementary fig. 6l–s). Furthermore, overexpression of ROCK2 in HK2-depleted TPC reduced the enhanced total tube length, branch points, and number observed in HUVEC co-cultured with HK2-depleted TPC (Supplementary fig. 6l–s). The above results showed that ROCK2, but not ROCK1, was the downstream effector of HK2 mediated pericyte's blood vessel supporting function.

Accumulating evidence suggest that tumor cells can secrete paracrine factors to affect their surrounding stromal cell metabolism[28]. To determine whether the difference in metabolic states between NPC and TPC was driven by tumor cells, we exposed NPC to conditioned medium from either NSCLC cell line A549 or HCC cancer cell line HepG2 and then used them for western blotting, glucose uptake, lactate concentration, and tube formation assays. Strikingly, exposure with the conditioned medium (CM) from cancer cells did not affect the glucose

uptake in NSCLC/HCC-derived NPC (Supplementary fig. 6t), while it increased the expression of HK2, ROCK2, and p-MLC2 as well as lactate production in NPC as compared with untreated NPC (Fig. 5p and Supplementary fig. 6u). When these CM treated cells co-cultured with HUVEC in a tube formation assay, it significantly decreased total tube length, branch points, and number as compared to HUVEC co-cultured with untreated NPC (Fig. 5q–t).

Recent studies suggested that tumor-derived hyaluronan (HA) fragments up-regulated glycolytic gene expression (including HK2) to modulate the pro-tumor function of monocytes, which could be rescued by using a HA antagonist (Pep-1)[29]. Interestingly, our flow cytometry analysis indicated that NSCLC/HCC-derived NPC expressed the HA receptor CD44 on their cell surface (Supplementary fig. 6v). We next examine the role of tumor cell-derived HA fragments in regulating pericyte metabolism by exposing NSCLC/HCC-derived NPC with conditioned medium from A549/HepG2 cells in the presence of HA antagonist (Pep-1) or control peptide (Cpep). Strikingly, our results showed that blockade of HA fragments by a HA antagonist (Pep-1) prohibited the up-regulation of HK2 and ROCK2 in NPC after treated with conditioned medium from A549/HepG2 cells as compared with the group treated with control peptides, while it had no effect on ROCK1 expression (Supplementary fig. 6w). To confirm ROCK2 as the downstream target of HK2 mediated glycolysis in TPC, we therefore pre-treated NSCLC/HCC-derived TPC with ROCK inhibitor, which were then used for lactate concentration measurement and seahorse assays. As expected, inhibition of ROCK did not affect the lactate concentration, ECAR and OCR in TPC as compared to placebo treated group (Supplementary fig. 7a–e), while treatment of ROCK inhibitor had no effect on HK2 expression in TPC (Supplementary fig. 7f, g). Overall, these data suggested that tumor-derived paracrine signal induced pericyte-HK2-driven glycolysis to dysregulate pericyte's blood vessel supporting role by up-regulating ROCK2-MLC2 mediated contractility.

**Administration of HK2 inhibitor enhances blood vessel function, doxorubicin delivery, and efficacy against tumor growth, while reducing hypoxia without affecting blood vessel number and coverage.** Previous studies from other laboratories and our

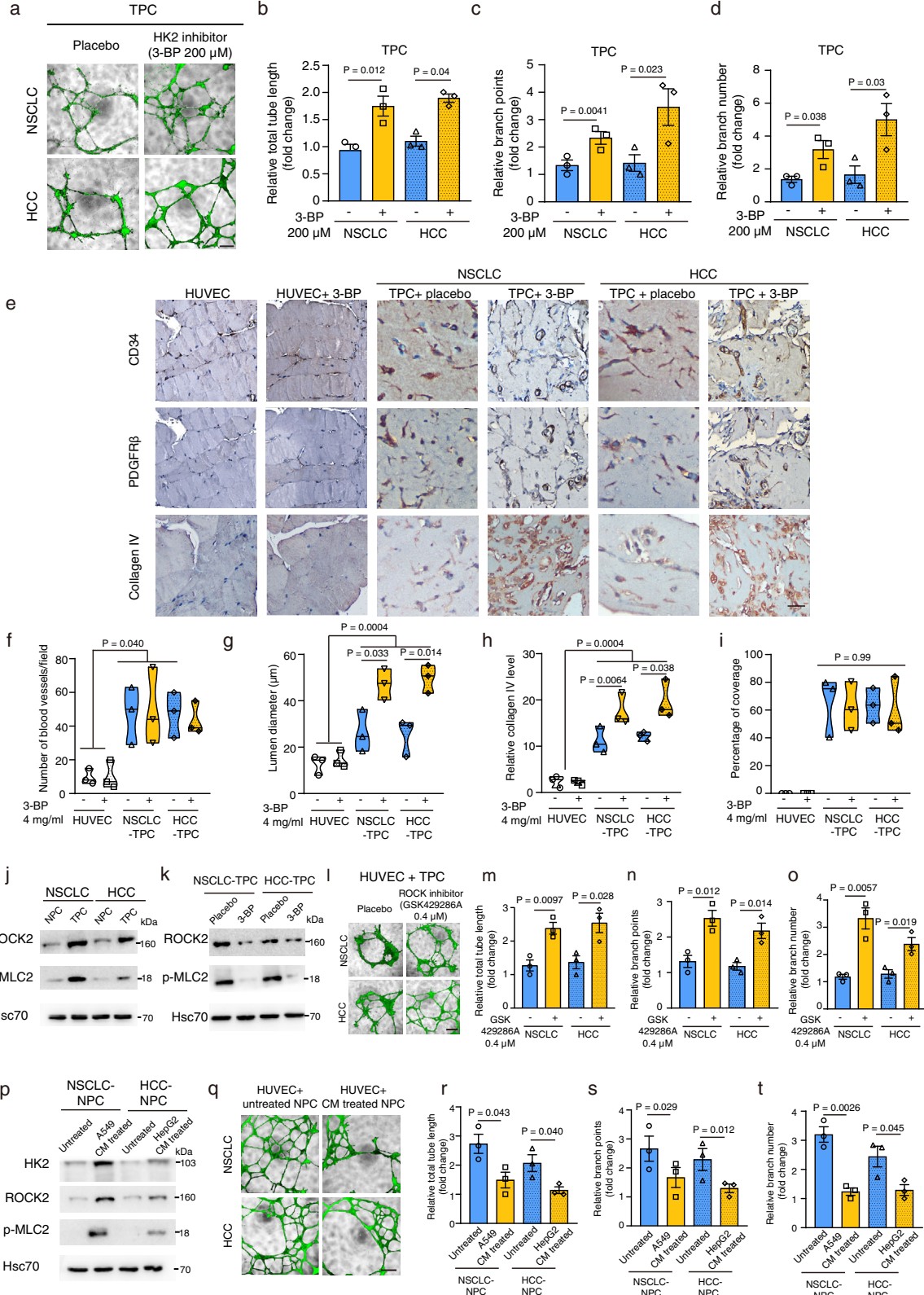

own indicated that improved tumor blood vessel functions can enhance chemotherapy delivery and efficacy against tumor growth and progression[16,30]. However, these works mainly focused on targeting endothelial cells, but not on pericytes. We next tested whether combination treatment with HK2 inhibitor and chemotherapeutic agent doxorubicin enhanced its efficacy against lung and liver tumor growth. Strikingly, our results

indicated that co-treatment with HK2 inhibitor and doxorubicin significantly inhibited the growth of subcutaneous lung (i.e. LLC, A549) or liver (i.e. mouse HCC cell line Hepa 1-6) tumors as compared to the group treated with either placebo, doxorubicin, or HK2 inhibitor alone (Fig. 6a–d and Supplementary fig. 8a–h). Notably, treatment with HK2 inhibitor or doxorubicin alone showed limited/modest effect on tumor growth as compared to

**Fig. 5 Expression of HK2 regulates the blood vessel supporting function of pericyte via ROCK2-MLC2 mediated contractility. a** Tube formation assay. HUVEC were co-cultured with CFSE labeled TPC pre-treated with either placebo or HK2 inhibitor (3-bromopyruvate, 3-BP). Representative immunofluorescent/bright field merged pictures of the tube formation after 24-h co-culturing in each group are given. **b–d** Bar charts show the relative total tube length, branch points, and number in each group ($n = 3$ independent experiments). Statistical tests were two-sided. **e** Representative IHC pictures of the serial sections of matrix plugs after immunostained with CD34, PDGFRβ or collagen IV antibody. **f–i** Matrix plug assay showed that treatment with HK2 inhibitor significantly enhanced the lumen diameter and collagen IV staining of blood vessels in HUVEC co-transplanted with TPC, compared to that treated with placebo. Violin plots show the relative number of blood vessels, lumen diameter, collagen IV level and percentage of coverage in each group ($n = 3$ mice per group). **j** Western blotting analysis of the ROCK2 and p-MLC2 expression in NSCLC/HCC-derived-NPC and -TPC ($n = 3$ independent experiments). **k** Administration of HK2 inhibitor 3-BP reduced the expression of ROCK2 and p-MLC2 in NSCLC/HCC-derived TPC ($n = 3$ independent experiments). **l–o** Tube formation assay of the HUVEC co-cultured with CFSE labeled NSCLC/HCC-derived TPC pre-treated with 0.4 μM ROCK inhibitor (GSK429286A) or placebo. Bar charts show the relative total tube length, branch points, and number in each group ($n = 3$ independent experiments). Statistical tests were two-sided. **p** Western blotting analysis of the HK2, ROCK2, and p-MLC2 expression in NSCLC/HCC-derived NPC after treated with conditioned medium (CM) from either A549 or HepG2 cells ($n = 3$ independent experiments). **q–t** Tube formation assay of the HUVEC co-cultured with untreated CFSE labeled NSCLC/HCC-derived NPC or NPC pre-treated with conditioned medium (CM) from A549 or HepG2 cells. Bar charts show the relative total tube length, branch points, and number in each group ($n = 3$ independent experiments). Statistical tests were two-sided. Results are given as means ± SEM. **b–d**, **m–o**, **r–t** Student's $t$ test. **f–i** One-way ANOVA. Scale bars in **a**, **l**, **q** represents 200 μm, **e** 50 μm.

placebo treated group (Fig. 6a–d and Supplementary fig. 8a–h). Analysis of lung tumor vasculature showed that any treatment including 3-BP had no significant effect on total blood vessel number and pericyte coverage as compared with other treatment groups (Fig. 6e–g and Supplementary fig. 8i–k), while it significantly enhanced tumor blood vessel diameter, basement membrane collagen IV staining, and blood vessel perfusion (Fig. 6h–j and Supplementary fig. 8l, m). Moreover, mice injected intravenously with Hoechst dye showed a significant increase in intratumoral Hoechst dye levels, relative to blood vessel number, after treatment with HK2 inhibitor alone, and with a combination of doxorubicin when compared with placebo or single doxorubicin treated mice (Fig. 6k). No significant difference in the body weight of LLC/A549 tumor-bearing mice during treatment was observed (Supplementary fig. 8n, o). Importantly, co-immunostaining analysis confirmed that LLC tumor pericytes expressed HK2 (Supplementary fig. 8p), while the expression of pericyte-p-MLC2 was reduced in LLC/A549 tumors after treated with HK2 inhibitor as compared with placebo (Supplementary fig. 8q). These results demonstrate that the pro-vascular functional effect of HK2 inhibitor was apparent in vivo, and corresponded with improved doxorubicin efficacy.

Since previous studies have shown a strong correlation between blood vessel perfusion/dilation and chemotherapeutic drug delivery[30,31], we next sought to examine whether the increase in blood vessel perfusion and dilation was sufficient to improve doxorubicin delivery in our animal models. Ex vivo two photon imaging showed that LLC tumor blood vessel perfusion, vascular diameter, and doxorubicin delivery were all increased in mice treated combination group as compared with doxorubicin alone treated groups (Fig. 6l–n and Supplementary movie 1, 2), implying that HK2 inhibitor treatment induced MLC2-driven tumor vasculature remodeling. Further confocal microscopy analysis confirmed that the intratumor doxorubicin level was increased in mice treated with HK2 inhibitor and doxorubicin when compared to mice treated with doxorubicin alone (Supplementary fig. 8r). Importantly, Doppler ultrasound analysis of microbubble perfusion revealed that treatment with HK2 inhibitor enhanced LLC tumor blood flow and perfusion in mice as compared with the groups treated with either placebo or doxorubicin alone (Fig. 6o, p). Notably, the increase in these dynamic flow and perfusion readouts was observed across the whole tumor, including the tumor core (Fig. 6o, p). Since tumor hypoxia has been linked with poor prognosis in cancer patients[30], we next examined the tumor expression of hypoxia indicator glucose transporter (Glut1) in all treatment groups, indicating that the HK2 inhibitor or HK2 inhibitor and doxorubicin

combination treatment significantly reduced tumor hypoxia (Supplementary fig. 8s), suggesting that the improved blood vessel function reduced tumor hypoxia in mice treated with HK2 inhibitor. To further examine whether the observed synergistic inhibitory effect of HK2 inhibitor and doxorubicin on tumor growth in vivo was due to direct inhibition on tumor cells or through modulating TPC function, we treated LLC cells with either placebo, 3-BP, doxorubicin or 3-BP, and doxorubicin combination in vitro, indicating that co-administration of 3-BP and doxorubicin had no synergistic inhibitory effect on LLC tumor cell growth in vitro (Supplementary fig. 8t). To determine whether 3-BP or/and doxorubicin could affect tumor cells mediated pericyte's glycolytic change, we examined the effect of conditioned medium harvested from cancer cells pre-treated with 3-BP or/and doxorubicin on pericyte-HK2 and ROCK2 expression. Both RT-PCR and western blot analysis indicated that the conditioned medium of 3-BP or/and DOX pre-treated tumor cells could still up-regulate HK2 and ROCK2 expression in NSCLC/ HCC-derived NPC as compared to untreated NPC group (Supplementary fig. 9a–e). Altogether, these data suggested that the synergetic inhibitory effect of 3-BP and doxorubicin on tumor growth in vivo was through modulating TPC's blood vessel supporting function.

To confirm the role of pericyte-HK2 in modulating tumor vasculature and subsequent drug delivery, we subcutaneously co-injected A549/MHCC-LM9 (i.e. human HCC cell line) cells with either stable HK2-depleted TPC, stable ROCK2 overexpressing HK2-depleted TPC or scramble transfected TPC at 10:1 ratio into nude mice and then treated the tumor-bearing mice with placebo or doxorubicin. Strikingly, administration of doxorubicin reduced tumor growth in the mice that were co-injected with A549/ MHCC-LM9 cells and HK2-depleted cells as compared to the mice that were co-injected with A549/MHCC-LM9 cells and scramble transfected TPC (Supplementary fig. 10a–h), while co-injection of A549/MHCC-LM9 cells with ROCK2 overexpressing HK2-depleted pericytes reduced the enhanced doxorubicin efficacy against tumor growth observed in mice co-injected with A549/MHCC-LM9 cells and HK2-depleted cells (Supplementary fig. 10a–h). No significance difference in total blood vessel number and pericyte coverage was observed between groups (Supplementary fig. 10i–k). Interestingly, the tumor blood vessel diameter, basement membrane collagen IV staining, blood vessel perfusion, and intratumoral Hoechst were all increased in mice that were co-injected with A549 cells and HK2-depleted TPC as compared to mice co-injected with A549 cells and scramble transfected TPC, while the increased blood vessel functions observed were reduced in mice that were co-injected with A549

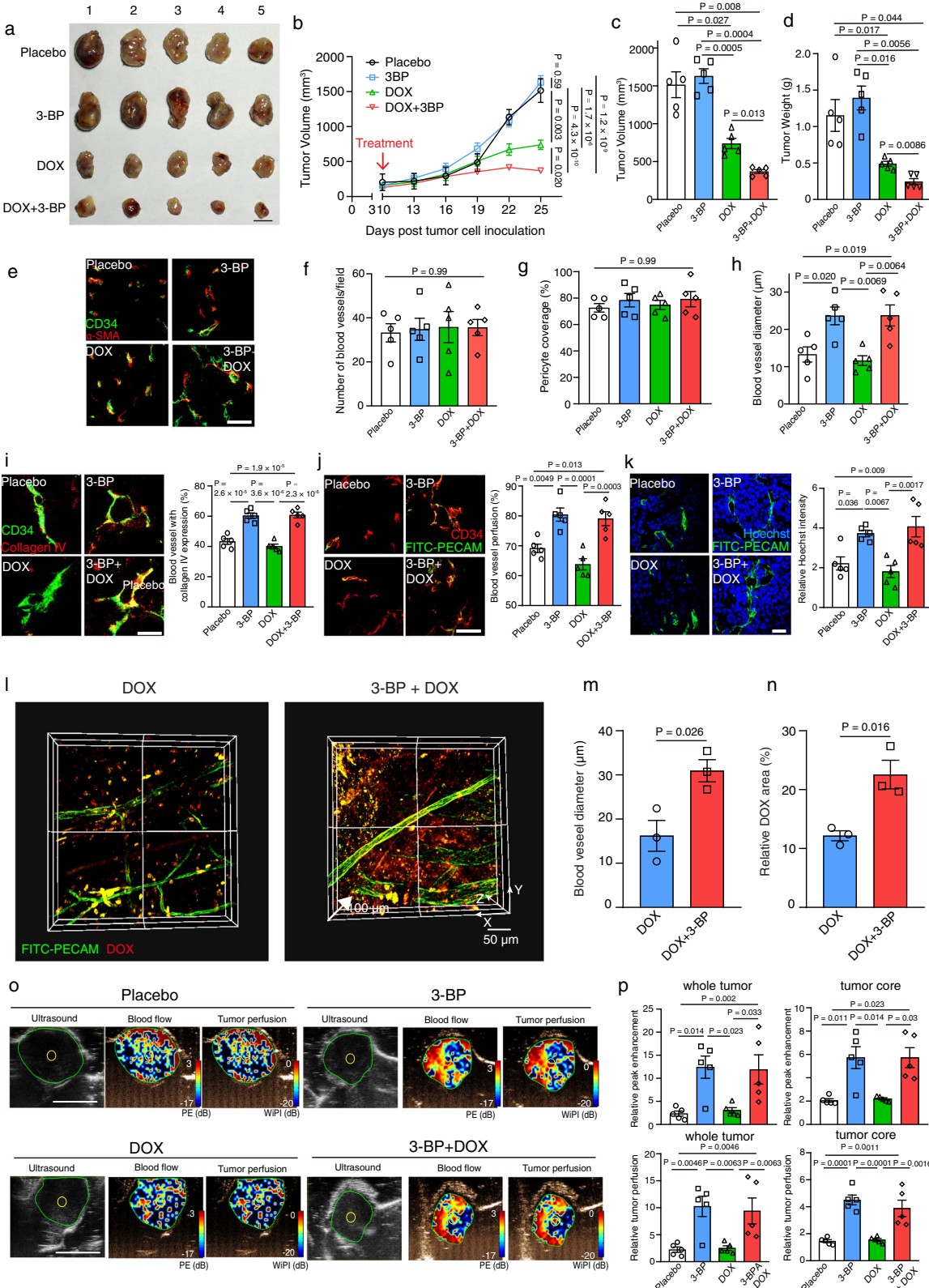

cells and ROCK2 overexpressing HK2-depleted TPC (Supplementary fig. 10l–o). Furthermore, the expression of pericyte-p-MLC2 was reduced in tumors arising from the mice that were co-injected with A549 cells and HK2-depleted TPC as compared to A549 cells co-injected with scramble transfected TPC, whilst its expression was increased in the tumors derived from co-injection of A549 cells and ROCK2 overexpressing HK2-depleted TPC

(Supplementary fig. 10p). Importantly, ex vivo two photon imaging showed that the tumor blood vessel perfusion, vascular diameter, and doxorubicin delivery were all increased in the mice that were co-injected with A549 cells and HK2-depleted TPC as compared with the mice that were co-injected with A549 cells and scramble transfected TPC, while the increased blood vessel perfusion, vascular diameter, and doxorubicin delivery observed

**Fig. 6 Administration of HK2 inhibitor remodels tumor vasculature to enhance blood flow, tumor perfusion, doxorubicin delivery, and efficacy against tumor growth. a** C57/BL6 mice were subcutaneously injected with $1 \times 10^6$ LLC cells and treated with placebo, 3-bromopyruvate (3-BP), doxorubicin (DOX), 3-BP, and DOX combination ($n = 5$ mice per group). Representative gross image of tumors from each treatment group is given. **b** Tumor growth curves showed that co-treatment of 3-BP and doxorubicin inhibited the growth of LLC tumors as compared with placebo, doxorubicin or 3-BP treated group. **c, d** Bart charts indicate the tumor weight and volume in each group. **e–i** Midline sections of LLC subcutaneous tumors were double immunostained for CD34 (green) and α-SMA/collagen IV (red) for blood vessel quantification, percentage of pericyte coverage, blood vessel diameter, and collagen IV intensity analysis. **j** LLC tumor bearing mice were intravenously injected with a FITC-PECAM antibody (green) to detect perfused blood vessels. **k** Bar charts show the relative area of Hoechst dye uptake into perivascular tumor cells to area of FITC-PECAM in each treatment group. **l–n** Ex vivo two photon microscopic imaging of FITC-PECAM antibody perfused LLC tumors after treated with DOX alone or 3-BP and DOX combination. Bar charts show the quantification of vessel diameter and doxorubicin in each group. Statistical tests were two-sided. **o–p** Doppler ultrasonography in live mice provides relative blood flow (peak enhancement, PE) and tumor perfusion (Wash-in perfusion index, WiPI) in subcutaneous LLC tumors. Representative ultrasound images. Bar charts, quantitation across the whole tumors, including tumor cores. Results are given as means ± SEM. **b** Two-way ANOVA. **c, d, f–k, p** One-way ANOVA. **m, n** Student's $t$ test. Scale bars in **a, o** represents 1 cm, **e, j, k, l** 50 μm, **i**, 20 μm.

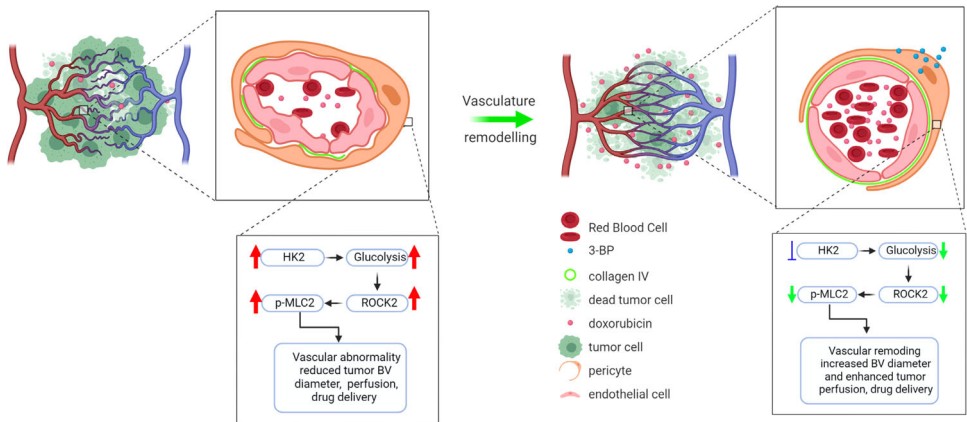

**Fig. 7 Schematic diagram represents the role of pericyte-HK2-driven glycolysis in regulating tumor blood vessel abnormality via ROCK2-MLC2 mediated contractility.** TPC up-regulates HK2-driven glycolysis to induce ROCK2-MLC2 mediated contractility, which in turn negatively regulates its blood vessel supporting role. Administration of HK2 inhibitor 3-BP induces MLC2-driven tumor vasculature remodeling to increase blood flow, tumor perfusion, blood vessel diameter as well as chemotherapeutic drug delivery and efficacy against tumor growth, while reducing tumor hypoxia without affecting blood vessel density and pericyte coverage.

was decreased in mice that were co-injected with A549 cells and ROCK2 overexpressing HK2-depleted TPC (Supplementary fig. 10q and Supplementary movie 3–5). Doppler ultrasound analysis of microbubble perfusion revealed that depletion of HK2 in TPC enhanced A549 tumor blood flow and perfusion in mice as compared with scramble control group (Supplementary fig. 10r), while overexpression of ROCK2 in HK2-depleted TPC reduced the enhanced tumor blood flow and perfusion observed (Supplementary fig. 10r). These results demonstrated the important role of pericyte-HK2-driven ROCK2-MLC2 mediated contractility in tumor vasculature remodeling and drug delivery.

Overall, our findings show that treatment with HK2 inhibitor or depletion of pericyte-HK2 by shRNA can enhance blood flow and perfusion leading to increased intratumoral drug delivery and efficacy against tumor growth, whilst decreasing tumor hypoxia without affecting blood vessel density and pericyte coverage (Fig. 7). These results indicate that targeting pericyte-HK2 expression/activity could be an effective treatment strategy for cancer patients.

## Discussion
Our method successfully purified a population of pericytes from normal adjacent tissues and tumors derived from NSCLC or HCC patients. We showed an elevated HK2-driven glycolysis in TPC as compared with NPC, which contributed to its abnormal blood vessel supporting function by activating ROCK2-MLC2 mediated contractility. Depletion of HK2/ROCK2 by shRNA or inhibitors reduces TPC contractility to remodel vasculature, while combined

treatment of HK2 inhibitor and doxorubicin enhanced the efficacy of chemotherapy drug against tumor growth in both lung and liver cancer animal models. This is an unexpected result and call for a reconsideration of our current strategy targeting pericytes for cancer treatment.

Pericytes, also known as contractile cells, play an important role to modulate angiogenesis, blood vessel diameter, blood flow, vascular perfusion, and integrity mainly via their contractile properties[1,7,10,32]. Interaction between pericyte-endothelial cell in the blood vessel wall is a fundamental process in regulating vascularization, vasoconstriction, vasodilation, and stabilization[9,14], while defective interaction between two cell types causes human pathological conditions, including dementia syndrome and tumor angiogenesis[1,12]. Indeed, genetic knock-out of PDGFB or PDGFRβ both causes vascular dysfunction and perinatal death[5]. Due to the lack of an effective human pericyte isolation protocol, most of the in vitro pericyte studies were performed by using either bovine/mouse retinal pericytes or mouse brain pericytes[4,18,33]. Although pericytes can potentially differentiate into other cell types[34], the retinal pericytes were neither isolated by FACS, nor fully characterized by pericyte marker expression or cultured in medium supplemented with pericyte growth factors in previous studies[32,35]. Furthermore, it remains debating about the mesenchymal stem cell property of pericytes in vivo[36]. Contrary to these studies, we developed a MACS method to enrich vascular cell population from the dissociated tissues and then subjected the cell population for subsequent FACS to purify pericytes, which were then cultured in commercially available medium dedicated for human pericytes. By

performing a series of characterization experiments, we confirmed the pericyte identity of our isolated cells derived from NSCLC and HCC, while there was no significant difference in basic function between NPC and TPC. In addition, our culturing method can maintain the pericyte identity of the isolated cells over passages.

We next sought to explore their roles in regulating vasculature in vitro and in vivo. Previously, Mcllroy et al.[37], show that exposure of immortalized HUVEC with bovine retina pericyte conditioned medium can inhibit its migration as compared with HUVEC exposed with serum-free medium or endothelial cell conditioned medium, while it also repressed the tube formation ability of HUVEC. On the other hand, Kumar et al.[38] found that mesenchymoangioblast-derived pericytes promoted blood vessel formation and stability in vitro and in vivo. The difference amongst these studies may be due to different sources/species of pericytes being used in the tube formation assays. Nevertheless, our data indicated that TPC exerted a stronger repressive effect on the total tube length, branch points, and number as compared with NPC, while in vivo co-transplantation of HUVEC with TPC formed smaller and irregular shaped blood vessels with poor basement membrane structure as compared with HUVEC co-transplanted with NPC. In addition, our cell contraction assays showed that NSCLC/HCC-derived TPC displayed a stronger carbachol-induced cell contraction as compared with NPC, fibroblast, or HUVEC. Overall, our study showed that TPC had defective blood vessel supporting functions as compared with NPC.

Although the contribution of specific metabolic programs to the regulation of cell state/functions have been extensively studied in endothelial cells, the role of metabolic state in regulating pericyte function remains unclear. Transcriptional profiling study showed a consistent increase in glycolytic gene hexokinase 2 expression in NSCLC/HCC-derived TPC as compared with NPC. Since HK2 plays a key role in catalyzing the first step of the glycolytic pathway where glucose is phosphorylated to glucose-6-phosphate[39], we demonstrated that the lactate production was up-regulated in TPC as compared with NPC, while seahorse assays showed an increase in ECAR but decreased OCR in TPC as compared that to NPC. Importantly, metabolic flux analysis indicated the relative abundance of lactate, but not the TCA cycle metabolites, was increased in TPC as compared to NPC. These findings were supported by the clinical observation that the pericyte expression of HK2 was up-regulated in tumors as compared with normal adjacent tissues, while high percentage of pericyte-HK2 positive blood vessels correlated with poor overall survival in NSCLC and HCC patients respectively. We next explored the role of pericyte-HK2 in regulating its blood vessel supporting function. Our results showed that pre-treatment with HK2 inhibitor enhanced the total tube length, branch points, and number in HUVEC co-cultured with TPC in a tube formation assay as compared with HUVEC co-cultured with placebo pre-treated TPC, while in vivo co-transplantation of HUVEC with HK2 inhibitor pre-treated TPC enhanced blood vessel diameter and collagen IV expression as compared with HUVEC co-transplanted with placebo-treated TPC. In addition, pre-treatment with HK2 inhibitor reduced carbachol-induced cell contraction in TPC as compared to placebo pre-treated TPC. Notably, treatment with HK2 inhibitor did not affect the tube formation ability of HUVEC on its own as well as the number of blood vessels in subcutaneous LLC and A549 tumors as compared with placebo treated group, suggesting that it has a specific role in pericyte's blood vessel supporting function. Additionally, our results highlight the potential role of pericyte contractility in regulating blood vessel function. Indeed, previous studies indicate that RhoA-ROCK (Rho-associated protein kinase) signaling pathway has an important role in regulating retinal pericyte

contractility[22,23], while Kutcher et al. show that pericyte contractility determines endothelial cell cycle progression and sprouting[22]. He et al.[40] has reported that targeting α-SMA+ pericyte contractility could normalize tumor vasculature in a spontaneous pancreatic cancer mouse model. In this study, we observed an up-regulation of ROCK2 expression and its downstream effector MLC2 phosphorylation in TPC as compared with NPC. while administration of HK2 inhibitor reduced ROCK2 and p-MLC2 expression in TPC. In addition, pre-treatment of ROCK inhibitor also decreased a carbachol-induced cell contraction in TPC as compared to placebo pre-treated TPC. Importantly, depleting HK2 or ROCK2, but not ROCK1, enhanced the total tube length, branch point and numbers in HUVEC co-cultured with TPC in tube formation assays as compared with HUVEC co-cultured with scramble transfected TPC, while overexpression of ROCK2 in HK2-depleted TPC reduced the enhanced effect observed. These results suggest that pericyte-HK2-driven glycolysis contributes to vascular dysfunction via activation of the ROCK2-MLC2 mediated contractility.

We next sought to explore the molecular control of HK2 expression in TPC. Since previous work showed that cancer cells can affect surrounding stromal cell metabolism via paracrine signaling[28], we therefore exposed NPC with conditioned medium from cancer cells, indicating that exposure of NPC with conditioned medium from cancer cells up-regulated HK2 and ROCK2 expression as compared to untreated NPC. Indeed, recent studies showed that tumor-derived HA fragments can activate glycolytic gene expression, including HK2, in monocytes[29]. Since NSCLC/HCC-derived NPC was shown to express the HA receptor CD44 in this study, we therefore examined its role in pericyte metabolic reprogramming, indicating that blockade of HA fragments prohibited the up-regulation of HK2 and ROCK2 expression in NPC after exposure with conditioned medium from cancer cells. Overall, our results show that tumor-derived paracrine signal up-regulates pericyte-HK2-driven glycolysis to dysregulate its blood vessel supporting role by up-regulating ROCK2 mediated pathway.

Cancer cells rely on glycolysis to meet their energy demand, which is known as the Warburg effect. Since HK2 is a key mediator of glycolysis, its inhibitor, 3-bromopyruvate, has been extensively tested as a single anticancer agent for treating preclinical tumor models[41,42]. Unfortunately, it has so far provided modest treatment outcomes in different types of cancers[43]. Indeed, we showed that treatment with HK2 inhibitor 3-bromopyruvate alone showed modest effect on subcutaneous lung or liver tumor growth, while co-administration of HK2 inhibitor and doxorubicin significantly inhibited the growth of the tumors by inducing MLC2-driven vasculature remodeling, while reducing tumor hypoxia without affecting tumor blood vessel density and pericyte coverage. By performing ex vivo two photon microscopy and Doppler ultrasound imaging, we proved that administration of 3-BP induced tumor vasculature remodeling to enhance blood flow, perfusion, and chemotherapy delivery. Interestingly, treatment with 3-BP and doxorubicin showed no synergistic inhibitory effect on LLC cell growth in vitro, while exposure of NPC with conditioned medium from 3-BP or/and doxorubicin pre-treated tumor cells could still up-regulate HK2 and ROCK2 expression as compared with untreated NPC. Together with the fact that the efficacy of chemotherapy delivery has been shown to be linked with tumor blood vessel perfusion and diameter[31], these findings suggest that the synergistic repressive effect of 3-BP inhibitor and doxorubicin on tumor growth in vivo relies on modulating pericyte's BV supporting role, but not through tumor cell inhibition. Additionally, the role of pericyte-HK2 in regulating MLC-2-driven tumor vasculature remodeling and drug delivery was further illustrated

by the co-injection experiments, showing that depletion of pericyte-HK2 expression also enhanced blood vessel perfusion and lumen diameter, which in turn increased chemotherapeutic drug delivery and efficacy against tumor growth, while over-expression of ROCK2 in HK2-depleted TPC reduced the enhanced blood vessel functions and drug delivery and efficacy against tumor growth observed in the mice that were co-injected with HK2-depleted TPC and cancer cells. Furthermore, triple immunostaining experiments indicated that the expression of pericyte-p-MLC2 was decreased in tumors arising from the mice that were co-injected with A549 and HK2-depleted TPC as compared to A549 co-injected with scramble transfected TPC, whilst its expression was enhanced in the tumor-derived from co-injection of A549 cells and ROCK2 overexpressing HK2-depleted TPC. Together, these results indicate that targeting pericyte-HK2-ROCK2-MLC2 mediated contractility improves blood vessel function and drug delivery and efficacy against lung and liver cancer growth.

Overall, our work develops a cellular model for studying human pericyte biology and shows that pericyte-HK2-driven glycolysis induces tumor blood vessel abnormality by activating ROCK2-MLC2 mediated contractility. Importantly, our data demonstrate, using strategy distinct from anti-angiogenesis or targeting pericyte coverage, that a radical approach of modulating the metabolic status of pericytes to remodel tumor vasculature for enhancing chemotherapy efficacy in cancer treatment.

## Methods

**Clinical specimens**. Human non-small-cell lung cancer and hepatocellular carci-noma paraffin-embedded samples and fresh tissue samples were obtained from Sun Yat-sen Memorial hospital with complete clinical data. The NSCLC patients and HCC patients who underwent curative freshly resected tissue enrolled for pericyte isolation were based on the following main criteria: (1) patients were absence of anticancer therapies prior to the operation. (2) No concurrent autoimmune disease, HIV, or syphilis. All human non-small lung cancer and hepatocellular carcinoma paraffin-embedded samples and fresh tissue samples were anonymously coded in accordance with local ethical guidelines (as requested by the Declaration of Hel-sinki) with written informed consent and a protocol approved by the review board of Sun Yat-sen Memorial hospital (Guangzhou, China).

**Correlation study between gene expression and patient survival using the Kaplan–Meier plotter database**. Both the RNA-seq data and clinical data including outcome and clinico-pathological information of human NSCLC and HCC patients were obtained from the Kaplan–Meier plotter database (KM plotter). All the survival studies were done according to the web tool developer's instruction (http://kmplot.com/analysis/)[44,45]. Briefly, the gene probes of our gene of interest were selected from the KM plotter database, which were then used to plot Kaplan–Meier survival curves in order to determine the correlation between their expression and patient survival in NSCLC and HCC, using auto best-selected cut-off or median setting. The relationship between the ratio of ROCK2 to PDGFRβ/CD146/ACTA2 expression and patient survival data was analyzed using the KM plotter database.

**Cell culture**. Human umbilical vein endothelial cells (HUVEC) were isolated and cultured in endothelial cell medium (ScienCell, 1001) according to the manu-facturer's instruction. Human primary dermal fibroblasts (HFF-1, SCSP-109) were purchased from National Infrastructure of Cell Line Resource and cultured in the Dulbecco's Modified Eagle Medium (DMEM) (GIBCO, C11995500BT), supple-mented with 10% fetal bovine serum (FBS) (Biological Industries, 04-400-1A). Human normal adjacent tissue derived pericytes and tumor-derived pericytes were isolated as follow and cultured in pericyte medium (ScienCell, 1201). Peripheral blood mononuclear cells (PBMC) were isolated from buffy coats derived from the blood of healthy donors by Ficoll density gradient. The human primary pulmonary artery smooth muscle cells (SMC, PCS-100-023) were purchased from ATCC and cultured in vascular cell basal medium (PCS-100-030, ATCC) supplemented with vascular smooth muscle growth kit components (PCS-100-042, ATCC). For HUVEC and pericytes culture, 0.1% gelatin (Sigma, 9000-70-8) containing 0.7% type I collagen (Sigma, C3867-1) was coating on plate or flask prior to seeing cells. Lewis lung carcinomas (LLC), A549, Hepa 1-6 and HepG2 cells (all from ATCC) as well as MHCC-LM9 cells (a gift from Professor Shimei Zhuang) were cultured in RPMI1640/DMEM medium supplemented with 10% fetal bovine serum and 1% penicillin and streptomycin.

**Purification of pericytes from normal adjacent tissues and tumors**. NSCLC and HCC specimens were used to isolate pericytes immediately after operation or preserved in MACS tissue storage solution (Miltenyi). Fresh tumor tissues or corresponding normal tissues were rinsed to remove the blood by phosphate-buffered saline (PBS) and the necrotic parts were also eliminated. Specimens were micro-dissected into small pieces of 0.5–1 cm$^3$ size. These pieces were digested by incubation for 45 min at 37 °C in shaking table with the digestive solution which composed of DMEM, supplemented with 1 mg/mL collagenase type 1 (Wor-thington, LS004194), 1 mg/mL collagenase type 3 (Worthington, LS004182), 1 mg/mL collagenase type 4 (Worthington, LS004182), and 1 mg/mL DNase I (Roche, 10104159). After the enzymatic digestion, the enzymes were discarded by cen-trifugation, the pellets were then resuspended and filtered through 100 μm and 70 μm strainers sequentially. The filtered cells were collected and red blood cells were then removed by using RBC Lysis Buffer (Biolegend, 420301). Afterwards, the dead cells and debris was removed by using dead cell removal kit (Miltenyi, 130-090-101) according to the manufacturer's instruction. The viable single cells were then immunostained with FITC-conjugated anti-human CD146 antibody (Biole-gend, 361012, 1:100), which were enriched by using the EasySep™ human FITC positive selection kit II (STEMCELL, 17662). The enriched CD146+ cells were subsequently immunostained with PE-conjugated anti-human CD45 (Biolegend, 368510, 1:100), PE/Cy7-conjugated anti-human CD31 (Biolegend, 303118, 1:100), APC-conjugated anti-human CD140b (PDGFRβ) (Biolegend, 323608, 1:100), and fixable viability dye (FVD) eFluor® 780 (ebioscience, 65-0865-14, 1:300). Finally, the immunostained cells were sorted under sterile condition using MoFlo Astrios EQ (Beckman Coulter) to purify FVD−CD31-CD45-CD146+CD140b+(PDGFRβ) pericytes.

**Characterization of pericytes by flow cytometric analysis**. Primary pericytes were immunostained with fluorochrome-conjugated antibodies of the FACS step, while an additional endothelial marker BV421-conjugated anti-human CD34 (Biolegend, 343610, 1:100), pericyte surface marker APC-conjugated anti-human CD13 (Biolegend, 301706, 1:300), or PE-conjugated anti-human CD44 (Biolegend, 338808, 1:100) was analyzed by performing flow cytometry.

**Immunohistochemistry and immunofluorescent staining**. Formalin-fixed, paraffin-embedded tissues were cut into 4-μm sections and subjected to Immuno-histochemistry (IHC) and Immunofluorescence (IF) staining. Antibodies used in the IHC and IF staining included: rabbit mAb against human CD34 (ZSGB-BIO, ZM-0046, 1:300), rabbit mAb against human/mouse CD34 (Abcam, ab81289, 1:300), mouse mAb against human CD146 (ZSGB-BIO, ZM-0299, 1:100), mouse mAb against human/mouse α-smooth muscle actin (α-SMA)-Cy3™ antibody (Sigma, C6198, 1:500), rabbit mAb against human PDGFRβ (CST, 3169S, 1:100), rabbit mAb against human/mouse Glut1 (Abcam, ab652, 1:100), rabbit mAb against human/mouse Hexokinase II (Abcam, ab227198, 1:100), rabbit mAb against human/mouse ROCK2 (Merck, HPA007459, 1:100), and rabbit mAb against human/mouse p-MLC2 (CST, 3671S, 1:100). Antigen retrieval was performed in the high pressure condition for 10 min within the unmasking buffer (Tris-HCL, pH = 9.2). For calculating the percentage of pericyte-HK2 positive blood vessels, the tumor sections from our NSCLC and HCC patient cohorts were triple immunostained with CD34, HK2, and α-SMA antibodies, which were then scanned by using the Pannoramic DESK scanner (3D HISTECH, Hungary). The percentage of HK2 and CD34 and α-SMA-triple-positive blood vessels was calculated as the number of HK2 and CD34 and αSMA triple-positive blood vessels over the total number of CD34 and αSMA-double-positive blood vessels. Patient data are expressed as those with less or more, than the mean percentage of blood vessels that are pericyte-HK2 positive (mean value for NSCLC: 47%; mean value for HCC: 48%).

For immunohistochemistry experiments, VECTASTAIN® Elite ABC-HRP Kit (VECTOR, PK-7200), and ImmPRESS® Excel Amplified Polymer Staining Kit (VECTOR, MP-7601-15) were used to detect human or mouse antigens respectively. For immunofluorescence, Alexa Fluor–conjugated secondary antibodies (Invitrogen Molecular Probes, A21202, A21206, A31570, A31572, and A21082, 1:500) were used to detect human antigens, and the sections were counterstained with 4′-6′-diamidino-2-phenylindole (DAPI, BD PharmingenTM, 564907). To apply the antibodies which were originated from the same host, Alexa Fluor™ 488 Tyramide Reagent (Invitrogen™, B40953) was used to remove the antibody but preserving the fluorescent signal.

**Coverslip cell staining**. Isolated pericytes (NPC, TPC), normal fibroblasts, HUVEC, and smooth muscle cells were first seeded onto coverslips in 12-well plates overnight, which were then fixed with 4% formaldehyde in PBS for 10 min at room temperature. Cells were blocked, permeabilized, and incubated with rabbit mAb against human FAP (E1V9V) (CST, 66562, 1:100), mouse mAb against human desmin (Merck Millipore, MAB3430, 1:100), or indicative mAbs. Alexa Fluor–conjugated secondary antibodies and DAPI were used as above.

**RNA extraction and real-time PCR**. Total RNA was extracted from the indicative cells using TRIzol reagent (Invitrogen, AM9738). Aliquots (2 μg) of total RNA were reverse transcribed into cDNA using 5× All-In-One RT MasterMix (Abm, G492) followed by quantitative real-time-PCR (qPCR) using HieffTM qPCR SYBR®

Green Master Mix (No Rox) (YEASEN, 11201ES08). All reactions were run in triplicate and performed on a Roche Light Cycler 480 Instrument II (Roche Diagnostics). The cycle threshold (Ct) values did not differ by >0.2 among the triplicates. The level of target genes was normalized to that of β-actin (internal control) to permit the calculation of the 2-ΔCt value. The primers used for qPCR are provided in Supplementary Table 1.

**Cell proliferation assay, doubling time measurement, and drug testing.** CCK-8 proliferation assay was done according to the (Jiangsu KeyGENBioTECH Corp., Ltd). Briefly, 1500 pericytes per well were seeded onto a 96-well plate and the cell proliferation was detected at indicated time points. For the conditioned medium experiment, 2000 HUVEC per well were seeded onto a 96-well plate. After 24 h, the cells were incubated with fresh endothelial cell medium containing conditioned medium from NPC/TPC (7 to 3 ratio). CCK-8 reagent was added to the culturing medium 4 h prior to each time point. Doubling time for pericytes at each passage was calculated by the formula: T (hours) = In (2) × duration of culture (hours)/In (output cell number/input cell number). For in vitro drug testing, 1500 LLC cells per well were first plated onto a 96-well plate overnight and treated with either 200 μM 3-BP, 2.5 μM doxorubicin or 3-BP, and doxorubicin combination for 48 h.

**Wound healing assay.** Pericytes or HUVEC were seeded onto 24-well plates at a density that reached 90% confluence after 24 h. A 200 μL pipette tip was used to starch a straight line across the wells. Afterwards, the HUVEC were incubated with fresh EC medium containing conditioned medium derived from NPC/TPC (7 to 3 ratio). Images were taken under bright field using a phrase contrast microscope (Nikon, Ni-U) at indicated time points. The cell migration was quantified by calculating the migrated area in each group.

**Trans-well migration assay.** An endothelial cell dependent recruitment model was developed by using a trans-well system. Briefly, $4 \times 10^4$ HUVECs were seeded onto a well of 24-well plate overnight, while $5 \times 10^3$ NPC or TPC were seed on the upper chamber of a trans-well insert in a 24-well plate overnight. Afterwards, the insert was then transferred to the 24-well plate containing HUVEC for 6 h. The migrated cells on the bottom part of the insert were then stained with crystal violet and counted by using a phrase contrast microscope.

**Senescent cell detection.** NPC and TPC at passage 3 and 12 were measured by using senescent cells histochemical staining kit (Sigma, CS0030). The pictures were captured by using a fluorescence microscope (Nikon, Ni-U).

**Cell viability/apoptosis assay.** Primary pericytes were harvested at 90% confluence density. Then the percentage of viable/apoptotic cells were evaluated by using FITC Annexin V apoptosis detection kit with PI (BioLegend, 6409914).

**Cell contraction assay.** Cell contraction assay was done by adding 1 mM Carbachol (Sigma, PHR1511) into 80% confluent monolayer of NPC, TPC, HUVEC, or fibroblast for 30 min. For the inhibition assay, TPC were pre-treated with 200 μM 3-BP or 0.4 μM GSK429286A inhibitors at room temperature for 15 mins. Time-lapse images were photographed by using an ImageXpress micro-confocal microscopy. ImageJ software was used to quantify the percentage of contractile cells in three different microscopic fields.

**Generation of HK2/ROCK1/ROCK2 stably depleted TPC and stable ROCK2 overexpressing HK2-depleted TPC.** shRNA specifically targeting human HK2/ROCK1/ROCK2 cloned into the psi-LVRU6P vectors were brought from Gene-Copoeia. A psi-LLVRU6P-scramble shRNA (GeneCopoeia) was used as a control. In addition, ROCK2 cDNA sequence was cloned into Lv105 vector was purchased from GeneCopoeia too. The recombinant construct was co-transfected with three lentivirus packing vectors, pLp1, pLp2, and pLp-VSVG (GeneCopoeia) into 293T cells (ATCC) using the Lipofectamine 3000 (Invitrogen). Viral supernatant was harvested from day 2–3, filtered with a 0.45 μm filter, which were then used to infect NSCLC/HCC-derived TPC in the presence of 6 μg/ml polybrene at 37 °C. After 48 h incubation, the culturing medium was replaced with fresh medium and selected with 2 μg/ml puromycin drug for 2 weeks. The selected cells were then used in the indicated experiments.

**Tube formation assay.** HUVEC were detached and resuspended in EGM™-2 MV Microvascular Endothelial Cell Growth Medium-2 BulletKit™ (Lonza, CC-3202) to a concentration of $2 \times 10^5$ cell/ml. NPC/TPC/fibroblast/smooth muscle cells or HK2/ROCK1/ROCK2 stably depleted NSCLC/HCC-derived TPC/ROCK2 overexpressing HK2 stably depleted TPC/scramble transfected TPC were first labeled with 5-(and -6)-carboxyfluorescein diacetate succinimidyl ester (CFSE) (TONBO, 13-0850), which were then co-cultured with HUVEC at ratio of 1:10 (NPC/TPC/fibroblast/smooth muscle cells vs HUVEC) on matrigel (R&D, 3432-010). In all, 20,000 HUVEC per well were seeded onto a 96-well plate. For the rescue experiments, TPC were first pre-treated with 200 μM 3-BP or 0.4 μM GSK429286A/HA-1077 inhibitors at room temperature for 15 min. The inhibitors

were then removed by centrifuging. For the conditioned medium experiment, NPC were first pre-treated with conditioned medium from A549 or HepG2 cells for 6 h. These cells were then co-cultured with HUVEC as described above. Images were taken by using a fluorescence microscopy 24 h after co-culturing (Nikon, Ni-U). The total tube length, branch points, and number were measured in each group.

**Matrix plug assay.** Matrix plug assay was done by mixing HUVEC with either NPC, untreated TPC, TPC pre-treated with placebo/HK2 inhibitor 3-bromopyruvate (3-BP), fibroblast, smooth muscle cells (2:1), which were then seeded in non-adherent round-bottom 96-well plates (Corning, 7007) and cultured at 37 °C overnight. All suspended cells contributed to the formation of a single complex spheroid. In all, 100 spheroids were then suspended in 500 μL matrigel (growth factor reduced; BD Biosciences) and fibrinogen (final concentration of 2 mg/mL; Sigma, F3879) containing 500 ng/mL vascular endothelial growth factor (VEGF, R&D, 293-VE) and basic fibroblast growth factor (bFGF, PeproTech, 450-33). After the addition of Thrombin (0.4 U; Calbiochem) to the mixture, the matrigel mixture was injected subcutaneously on the lateral to abdominal midline region of 8-weeks-old nude mice. For the HK2 inhibitor experiment, the mice bearing implants were administered with an intraperitoneal injection of 40 mg/kg 3-BP twice a week until retrieved the implants. The implants were then formalin-fixed, embedded, and sectioned, while the serial sections were immunostained with anti-human CD34, PDGFRβ, and collagen IV (Abcam, ab6586). The blood vessel density, diameter, percentage of coverage, and collagen IV staining in each group were quantified by counting 5 microscopic fields using ImageJ software.

**Western blotting.** Cells were harvested and lysed with NP40 lysis buffer (Invitrogen). Protein concentration was determined by using the Bio-Rad Dc protein assay kit (Bio-Rad Laboratories). In total, 20–30 μg of protein from each sample was loaded onto 8–12% polyacrylamide gels. The protein was transferred to a nitrocellulose membrane and incubated with 5% milk in phosphate buffered saline with 0.1% Tween-20 (PBS-T), followed by an overnight incubation of primary antibody diluted 1 in 1000–2000 in 2% milk PBS-T at 4 °C. The blots were then washed three times with PBS-T and incubated with the relevant horseradish per-oxidase (HRP)-conjugated antibody diluted 1:1000 in 2% milk in PBS-T, for 1 h at RT. Chemiluminescence was detected by using Mini Chemiluminescent Imaging and Analysis System (Sagecreation). Rabbit mAb against human ROCK1 (CST, 4035, 1:1000) and rabbit mAb against human/mouse ROCK2 (CST, 8236S, 1:1000) were used in this part. Hsc70 was used as loading control (Santa Cruz, sc-7298, 1:1000). Primary antibody details were mentioned in the immunohistochemistry section. For the inhibitor experiment, NPC/TPC were first treated with either placebo, 3-BP, ROCK inhibitor for 24 h before harvesting for western blotting. The uncropped and unprocessed scans of all western blots were given in the Source Data File.

**Hyaluronan blocking experiment.** The HA antagonist Pep-1 (GAHWQF-NALTVR) and a control peptide (Cpep) (WRHGFALTAVNQ) were synthesized by GL Biochem (Shanghai, China)[46]. For the HA blocking experiment, NSCLC/HCC-derived TPC were treated with conditioned medium derived from A549/HepG2 cells in the presence of either 100 μg/ml HA antagonist Pep-1 or control peptide for 20 h before harvesting for western blotting analysis.

**Glucose uptake assay.** In total, $3 \times 10^4$ NPC or TPC/well in 100 μl culture medium were seeded onto a 96-well plate and grown overnight. The next day, the cells were treated with or without the indicative inhibitors for 15 mins. Then cells were washed and cultured with glucose-free medium for 4 h. In all, 20 min before flow cytometric assay, 100 μg/ml 2-NBDG was added into the culturing medium. In some experiments, NPC were incubated with fresh PC medium containing conditioned medium derived from tumor cells (7 to 3 ratio) for 12 h. Then these cells were performed as above. The cellular glucose level was detected by using Glucose Uptake Cell-Based Assay Kit (Cayman, 600470) according to the manu-facturer's instruction.

**L-lactate detection assay.** In total, 3000 NPC or TPC were plated into a 96-well plate and incubated with 120 μl of pericyte medium overnight. The next day, the cells were treated with or without the indicative inhibitors for 15 mins at room temperature. The cells were then washed and cultured with serum free medium for 12 h. In some experiments, NPC were incubated with fresh PC medium containing conditioned medium derived from tumor cells (7 to 3 ratio) for 12 h. Then these cells were performed as above. The L-lactate concentration of culturing medium was detected by using a glycolysis cell-based Assay Kit (Cayman, 600450) according to the manufacturer's instruction.

**Proteomics analysis.** For the sample preparation, NPC and TPC were cultured in a 100-mm culture plate at 37 °C in a 5% $CO_2$ waterlogged atmosphere using pericyte medium. Confluent cells were harvested by scraping with 800 μl of RIPA buffer. Cell lysates were centrifuged (4 °C 30 min, 15,000×g), the supernatant was collected and its protein content was determined using the BCA Protein Quanti-fication Kit (ThermoFisher Scientific). Pericyte protein extracts were sequentially

digested in-solution with trypsin. Briefly, the samples were initially precipitated with acetone, then resuspended with 50 Mm urea buffer, reduced with dithiothreitol (DTT, 2 mM, 37 °C 1.5 h), alkylated with iodoacetamide (IAM, 10 mM, 25 °C 40 min), and they were diluted to 60 mM urea for overnight digestion with trypsin (Sequencing Grade Modified Trypsin, V5111, Promega) at 37 °C. Samples were then diluted 2-fold again and digested overnight with trypsin at 37 °C Tryptic peptides were desalted using a HLB column (Oasis HLB, 186000383, Waters), evaporated to dryness. In all, 300 μg of dried pericyte protein digest were dissolved with 10 μl of 0.1% formic acid with H2O)[47].

The mass spectrometer was operated in positive ionization mode with an easy nano flex (Nanospray ionization, NSI) with spray voltage set at 2.3 kV and source temperature at 320 °C. The instrument was operated in data-dependent acquisition mode, with full MS scans over a mass range of m/z 300–2000. In each cycle of data-dependent acquisition analysis, following each survey scan, the most intense ions above a threshold ion count of 2.2e4 were selected for fragmentation at normalized collision energy of 30% (HCD). The number of selected precursor ions for fragmentation was determined by the "Top Speed" acquisition algorithm and a dynamic exclusion of 60 s. Fragment ion spectra were acquired in the linear ion trap (IT) or the Orbitrap (OT,60 K resolution) depending on the method, with an AGC of 4.0e5 and a maximum injection time of 50 ms for ion trap MS2 detection, and an AGC of 5.0e4 and a maximum injection time of 50 ms for Orbitrap MS2 detection. All data were acquired with Xcalibur software v4.0. Pericyte peptide was injected into a nano-UPLC system (EASY-nLC 1200 liquid chromatography) equipped with a 50-cm C18 column (EASY-Spray; 75-75-mn (PepMap RSLC C18, 2-6 μR particles, 45 °C) and hyphenated to an Thermo Scientific Orbitrap Fusion Tribrid Mass Spectrometer. Chromatographic gradient lengths 180 min were tested for peptide separation. Mobile phase: A, H2O: HCOOH (100: 0.01); B, ACN: H2O: HCOOH (80: 20: 0.01). Gradient elution: 0–5 min, 4–8% B; 5–152 min, 8–28% B; 152–172 min, 28–38%B; 172–175 min, 38–100%B; 175–180 min, 100%B, followed by equilibration of the column with mobile phase A as described previously[48].

Proteome Discoverer software suite (v2.3, Thermo Fisher Scientific) was used for peptide identification and quantitation. The data was searched against the Swiss-Prot human database. At the MS1 level, a precursor ion mass tolerance of 7 ppm was used, and up to three missed cleavages were allowed. The fragment ion mass tolerance was set to 20 ppm for the Orbitrap MS2 detection methods and to 0.5 Da for the linear ion trap MS2 detection methods. Oxidation of methionine, and N-terminal protein acetylation were defined as variable modifications whereas carbamidomethylation on cysteines was set as a fixed modification as described previously[48].

**Metabolite extraction and derivatization**. Tracer chemically defined medium was first prepared for metabolic flux analysis, which consisted of glucose-free DMEM medium (Gibco, 11966025) supplemented with [13 C] glucose (Cambridge Isotope Laboratories, CLM-1396-1). Pericytes were seeded onto six-well plate and incubated with the tracer chemically defined medium at 37 °C for 24 h. Afterwards, the metabolite extraction was done as described previously[49]. Polar metabolites (including organic acids and sugars) were extracted with methanol/water/chloroform. Briefly, spent medium was removed, and cells were rinsed with 0.9% (w/v) saline, and 250 μl of −80 °C methanol was added to quench metabolism. In total,100 μL of ice-cold water containing 1 mg norvaline internal standard was added to each well. Both solution and cells were collected via scraping. Cell lysates were transferred to fresh tubes, and 250 μL of −30 °C chloroform sample tubes, vortexing and centrifugation, the top aqueous layer (polar metabolites) and bottom organic layer were removed and dried under airflow. Derivatization of polar metabolites was performed using the gerstel multi-purpose sampler (MPS 2XL). Dried polar metabolites were dissolved in 20 mL of 2% (w/v) methoxyamine hydrochloride (MP Biomedicals) in pyridine and held at 37 °C for 60 min. Subsequent conversion to their tert-butyldimethylsilyl (tBDMS) derivatives for organic acids was accomplished by adding 30 mL N-methyl-N-(tert-butyldimethylsilyl) trifluoroacetamide supplemented with 1% tert-butyldimethylchlorosilane (Regis Technologies) and incubating at 37 °C for 30 mins.

**Gas chromatography/mass spectrometry analysis**. GC/MS analysis was performed using a Thermo 1300 equipped with a DB-35MS capillary column (30 m, 0.25 mm i.d., 0.25 μm coating) and connected to a Thermo ISQ MS. GC/MS was operated under electron impact ionization at 70 eV. One microlitre of sample was injected in split less mode at 270 °C, using helium as the carrier gas at a flow rate of 1.2 mL/min. For measurement of organic and amino acids, the GC oven temperature was held at 80 °C for 2 min and increased to 255 °C at 3.5 °C/min, and then ramped to 320 °C at 15 °C/min for a total run time of ~60 mins with 5 mins of delay time. The MS source was held at 300 °C, and the detector was run in scanning mode, recording ion abundance in the range of 100–650 m/z.

**Metabolite quantification and isotopomer spectral analysis**. All results were based on chromatographic peak areas that received from targeted data evaluation in the Tracefinder 4.1 (Thermo Scientific) and analyzed as described previously[50]. Slight smoothing as implemented in peak detection by Tracefinder (5 points) was applied to improve automated peak integration. Mole percent enrichment (MPE) of isotopes was calculated as the percent of all atoms within the metabolite pool that were labeled.

**Measurement of ECAR and OCR by Seahorse analyser**. Seahorse assays were performed according to the manufacturer's instructions. Briefly, $2.4 \times 10^5$ cells/mL NPC or TPC were seeded onto a 96-well Agilent Seahorse XF cell culture microplates (Agilent) and incubated with human pericyte medium. For the ROCK inhibitor experiments, NSCLC/HCC-derived TPC was first pre-treated with 0.4 μM ROCK inhibitor GSK429286A for 1 h and then washed off before carrying on the ECAR and OCR measurement. A standard manufacturer-recommended two-step seeding procedure was utilized. After achieving cell adherence, the microplates were placed overnight in a 37 °C, 5% $CO_2$ incubator. Afterwards, the incubation medium for each well was aspirated, while the adherent cells were washed and then incubated with the Seahorse XF Base Medium (w/o phenol red) supplemented with 1 M glutamine (final concentration of 1 mM), 1 M glucose (final concentration of 10 mM) of OCR, 1 M glutamine (final concentration of 1 mM) of ECAR (adjusted the pH to 7.4 with 0.1 M NaOH) at a 37 °C, non-$CO_2$ incubator for 1 h before being transferred onto the microplate stage of a Agilent Seahorse XF96 flux analyser (Agilent). OCR and ECAR readings were taken using a 2-mins mix, 1-min wait, and 2-mins read cycling protocol. Various compounds were injected to alter the metabolic environment or to help determine the effects of an altered metabolic environment. These compounds included oligomycin, carbonyl cyanide-ptrifluoromethoxyphenylhydrazone (FCCP) and rotenone/antimycin of OCR, and glucose, 2-deoxyglucose (2-DG), and oligomycin of ECAR. OCR-ECAR data are presented as absolute rates or as percent changes from a stable baseline. For comparisons of data points between cells maintained under different conditions, group means were compared by Student's t-tests, with P-values <0.05 considered significant. All wells were normalized with protein concentration by using BCA assay.

**Conditioned medium experiment**. NSCLC/HCC-derived NPC was exposed with fresh PC medium containing conditioned medium derived from A549/HepG2 cells (7 to 3 ratio) for 6 h, which were used for western blotting and RT-PCR analysis. For the metabolic study, A549/HepG2 cells were first pre-treated with either placebo, 200 μM 3BP, 2.5 μM doxorubicin or 3BP and doxorubicin combination for 8 h and then washed off and incubated with fresh culturing medium for 16 h. The conditioned medium was then harvested, which were then mixed with fresh pericyte medium (7 to 3 ratio) and subsequently incubated with NSCLC/HCC-derived NPC for 24 h before harvesting for RT-PCR and western blotting analysis.

**Mouse tumor models**. In all, 4–6-weeks-old C57/BLK6 mice or nude mice were subcutaneously injected with either $1 \times 10^6$ LLC cells, $3 \times 10^6$ A549 cells, or Hepa 1–6 cells into the flank for subcutaneous tumor growth. For the co-injection experiments, the nude mice were co-injected with $3 \times 10^6$ A549/MHCC-LM9 cells and either $3 \times 10^5$ HK2-depleted TPC, scramble transfected TPC or ROCK overexpressing HK2-depleted TPC subcutaneously (10:1 ratio). Tumor growth was measured by using callipers every 3 days. Tumor volume (V) was calculated: $V = (length \times width^2) \times 0.52$. Once the tumor size reached 100 mm³, the tumor-bearing mice were treated with either placebo, HK2 inhibitor 3-BP (4 mg/kg, intraperitoneal Injection (IP)), doxorubicin (4 mg/kg, intravenous injection (IV)), and HK2 inhibitor (3-BP) and doxorubicin combination via IP/IV injection every 3 days for up to 15–18 days. When the treatment finished, animals were culled, tumor excised and either fixed in 4% formaldehyde in PBS overnight, or snap frozen in liquid nitrogen for subsequent immunostaining analysis. All animal procedures were approved by the Institutional Animal Care and Use Committee (IACUC) of Sun Yat-sen University.

**Blood vessel density and pericyte coverage**. The number of CD34-positive blood vessels per field in tumor sections from each treatment group was counted. For pericyte coverage, the percentage of α-SMA and CD34-double positive blood vessels over the total number of CD34-positive blood vessels were counted.

**Vascular perfusion**. Vascular perfusion was visualized by injecting LLC tumor-bearing mice via the tail vein with FITC-conjugated 20 μg of rat monoclonal anti-PECAM (BioLegend-#102406) 10 min and Hoechst (0.4 mg, Sigma Aldrich) 1 min prior to culling, respectively. Tumors were snap frozen, sectioned, and immunostained for CD34. The ratio (%) of double-positive blood vessels over CD34-positive vessels provided an indication of blood vessel perfusion.

**Blood vessel collagen IV assessment**. The percentage of CD34-positive vessels with collagen IV expression was calculated as the number of collagen IV and CD34-double positive blood vessels over the total number of CD34-positive blood vessels.

**Hypoxia quantification**. For Glut1 staining quantification, tumor sections were stained and analyzed using confocal microscopy. The staining intensity from 5 different fields was calculated by ImageJ software.

**Doxorubicin quantification by confocal microscopy**. For doxorubicin quantification, tumor-bearing mice were injected with 4 mg/kg doxorubicin 1 h prior to culling. The tumors were harvested and snap-frozen and the tumor sections were fixed and imaged by using confocal microscopy. The fluorescence signal of DOX from five different fields was analyzed by ImageJ software.

**Ex vivo two photon microscopic imaging**. For the quantification of doxorubicin and vascular perfusion, LLC tumor-bearing mice were injected with 4 mg/kg doxorubicin 1 h and 20 μg of rat monoclonal anti-FITC-PECAM antibody 10 min prior to culling, respectively. Tumor were harvested and then imaged by using the 25 × water immersion objective (Olympus, NA 1.05) of a confocal microscope (Olympus) equipped with a tunable pulsed chameleon infrared multiphoton laser (MaiTai eHODS and INSIGHT X3) and PMT (GaAsp). Z-stack images were taken up to 100 μm with 191 slices.

**Ultrasound analysis of blood flow and tumor perfusion**. High-resolution ultrasound imaging of LLC tumors using the Vevo® 2100 system with a MS250 transducer was performed according to the manufacturer's instruction (Visualsonics). For blood flow and tumor perfusion measurement, contrast-enhanced ultrasound imaging was carried out according to manufacturer's instructions (Vevo® 2100 system). Briefly, LLC tumor bearing mice were administered with 100 μL of bolus Vevo Micromarker suspension (VisualSonics) via tail vein using a tail vein Micromarker™ catheter. The ultrasound images were processed and analyzed by using VevoCQ™ contrast quantification software according to the manufacturer's instruction.

**Quantification and statistical analysis**. Student's $t$ test, Log-rank (Mantel-Cox) test and one/two-way ANOVA were performed for the statistical analysis and the details can be found in the figure legends: test used, exact value of n and dispersion measures. Statistical analysis was performed using GraphPad Prism.

**Reporting summary**. Further information on research design is available in the Nature Research Reporting Summary linked to this article.

## Data availability

The mass spectrometry proteomics data generated in this study has been deposited to the Proteome X consortium via the PRIDE partner repository with the dataset identifier PXD026963. The source data underlying Figs. 1c, f, k, l, 2c–g, 3a, b, d–f, h–k, 4b, d–k, 5b–d, f–k, m–p, r–t, 6b–d, f–k, m, n, p and Supplementary Figs. 3c–i, 4b–d, f, 5a–e, 6a–d, f–k, m–o, q–w, 7a–g, 8b–d, f–h, j–o, r–t, 9a–e, 10b–d, f–h, j–o, q, r are provided as source data. The use of publicly available data from NSCLC and HCC were consulted on the websites: https://kmplot.com/analysis/index.php?p=service&cancer=lung; https://kmplot.com/analysis/index.php?p=service&cancer=pancancer_rnaseq), under the specific product names: KM Plotter-Lung Cancer and-Pan-cancer RNA-seq. All other relevant data supporting the key findings of this study are available within the article and its supplementary information files or from the corresponding author upon reasonable request. A reporting summary for this article is available as a supplementary information file. Source data are provided within this paper. Source data are provided with this paper.

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

## Acknowledgements

The authors thank Professor Kairbaan Hodivala-Dilke for her technical advice. Schematic diagrams were created with BioRender.com. This work was supported by the National Natural Science Foundation of China (81920108028, 81872142, and 81902899); Guangzhou Science and Technology Program Key Project (201904020008); Guangdong Science and Technology Department (2020A0505100029, 2020B1212060018, and 2020B1212030004); and Guangdong Natural Science Foundation (Grant no. 2019A1515011802 and 2020A1515011280).

## Author contributions

Y-M.M., X.J. and X.Z. carried out majority of the experiments and contributed equally to the paper; Q.M., S.W., Y.C., X.K., X.Q., L.S. and C.H. assisted with the experiments; M.W. and C.L. provided clinical samples; P.-P.W conceived the study, supervised the research, and wrote the manuscript.

## Competing interests

The authors declare no competing interests.
