## [Peer Review File · Nature Communications]

Mural cell-HK2 driven glycolysis induces tumor blood vessel abnormality by activating ROCK2-MLC2 mediated contractilityREVIEWER COMMENTS

Reviewer #1 (Remarks to the Author); expert on pericytes and endothelial cells:

Meng et al have developed a smart and elegant strategy for the isolation and culture of human-derived pericytes from normal and tumour lung and liver specimens. An approach which is lacking in the field, and it will definitely have a strong impact. They have characterized the biology of what they define as “normal” pericytes and “tumor” pericytes and their effect on angiogenesis in vitro and in vivo. They claim that HK2-driven glycolysis in cancer pericytes leads to excessive pericyte contractility, what precludes correct vessel functioning in tumors. By pharmacologically inhibition of HK2, they show an improvement of tumor perfusion and an enhanced effect of chemotherapy.

Overall, this manuscript is experimentally well performed, and the data are convincing. However, it often seems a bunch of interesting observations which lack a clear message/flow. Some of the parts could be simply improved by a making an effort with the wording and the rationale of the experiments. But in many others, authors should make an effort strengthen the data.

1. How do authors explain that high HK2 expression reduces vessel diameter? They claim that that HK2 regulates cell contractility via ROCK expression, but this is hardly convincing from the data provided.

2. CD146 staining seems very weak, especially in NSLCC, at least in the pictures showed in Figure 1B. Only the depicted big vessel in the inset seems to be positive for CD146 (yellow), the other two are only red (α SMA positive) and the majority of the small vessels are only CD34 positive. Given that the MACS enrichment strategy is based on CD146 expression in vascular cells, this picture generates doubts about the percentage of pericytes obtained after FACS purification (Figure 1F and H). Also, this percentage is almost double in tumor samples compared with normal tissue. More images and a quantification of CD146 expression in tissue sections of normal and tumoral section will help to clarify this issue.

3. It seems that pericytes in vitro behave quite similarly irrespective of the origin. Are isolated pericytes losing their properties/dedifferentiating when cultured? However, when combined with endothelial cells, TPC seem to enhance tube/EC contraction compared to NPC. Again, the data are not convincing. How do authors explain that TPC express higher HK2 levels?

4. Figure 4J. Authors claim that HK2 expression is significantly increased in mural cells in tumors. Quantification of that expression in representative images should be provided. Endothelial cells seem to be contributing to HK2 overexpression too, as well as many other tumoral/stromal cells. This observation compromises the pericyte-driven hypothesis of the effects of in vivo, pharmacological inhibition of HK2. Pre-treated or genetically modified pericytes should be implanted together with tumor cells or, ideally, tumor cells should be implanted in pericyte-specific HK2 deficient mice.

5. Conclusions obtained from clinical correlation data (Figure 4K, Suppl Fig 4E) on HK2 expression are not valid, markers used are not specific of pericytes. PDGFR β and α SMA are expressed by cancer fibroblasts and CD146 is also expressed in endothelial cells.

6. Data on PDGFR β expression in cultured cells in Figure 2B does not match the qPCR profiles, as fibroblasts in the IF image show much greater expression than qPCR graph depicts compared with pericytes

Reviewer #2 (Remarks to the Author); expert on metabolism:

Pericytes play an important role in angiogenesis and tumor microenvironment formation. The authors developed a novel method to isolate pericytes from human normal and tumor tissues. This is a robust model for studying the role of pericytes in angiogenesis and cancer progression. The authors

discovered that pericytes from tumor had higher glycolytic flux but lower mitochondrial activity, suggesting that the TPC also display the Warburg effect, which alters cell contractility and limit blood supply. This could further exacerbate tumor hypoxia.

A few additional experiments that could strengthen the conclusions:

In 125-128, how did the authors conclude that 'CD146 was expressed in both endothelial cells and pericytes ... , while CD34 and PDGFR β was specifically expressed in endothelial cells and pericytes respectively'? Were independent markers used to confirm those cells are endothelial cells and pericytes respectively? How can they tell that CD146 was expressed in pericyte without showing the co-staining of CD146 and PDGFR β in Figure1?

The data demonstrated a strong association between HK2/glycolysis upregulation and ROCK2 activation. However, it is not clear whether the paracrine factors from tumor cells reprogram metabolism to activate ROCK2, or activate ROCK2 to reprogram metabolism. Does ROCK2 inhibitor represses the Warburg effect in TPCs?

3BP and Dox had synergistic effect in repressing tumor growth in vivo. It is difficult to dissect whether it was due to direct inhibition of tumor cells, or through modulating TPC functions. Does 3BP and Dox combination inhibit LLC cell growth in vitro? Can the CM from 3BP and Dox treated tumor cells activate HK2 and ROCK2 in pericytes?

Reviewer #3 (Remarks to the Author); expert on ROCK2:

This paper nicely demonstrates the distinct differences between normal and tumour associated pericyte function in angiogenesis and demonstrates the use of new isolation protocol to facilitate a deeper understanding of the differences between their signalling in vitro and recapitulated in vivo. Moreover, they reveal distinct differences in glycolysis and the role the ROCK2-MLC2 signalling plays in vasculature normalisation and importantly capitalise upon this to show that manipulation of this can improve chemotherapy delivery.

Can the authors perform some shRNA experiments of manipulation to show ROCK 2 effect are key to this work rather than ROCK1 off target effect- this should be easy enough to perform to demonstrate that ROCK2 rather than pan drugs such as GSK429286A are having a role in improving chemotherapy delivery beyond off target ROCK1 vascular relaxation effects?

Any other rock2 specific manipulation to address the above point would also satisfy this issue.

Lastly can the authors perform experiments that the effect they see in vivo for LLC (ie improve chemotherapy response) also hold true for HCC cancer as this is a major (ie half of the paper) aspect of this work which is not show as a final in vivo experiment?

REVIEWER COMMENTS

Reviewer #1 (Remarks to the Author); expert on pericytes and endothelial cells:

Meng et al have developed a smart and elegant strategy for the isolation and culture of human-derived pericytes from normal and tumour lung and liver specimens. An approach which is lacking in the field, and it will definitely have a strong impact. They have characterized the biology of what they define as “normal” pericytes and “tumor” pericytes and their effect on angiogenesis in vitro and in vivo. They claim that HK2-driven glycolysis in cancer pericytes leads to excessive pericyte contractility, what precludes correct vessel functioning in tumors. By pharmacologically inhibition of HK2, they show an improvement of tumor perfusion and an enhanced effect of chemotherapy.

We would like to thank for the reviewer’s positive support about the impact of our strategy for the isolation and culture of human-derived pericytes from normal and tumor lung and liver specimens.

Overall, this manuscript is experimentally well performed, and the data are convincing. However, it often seems a bunch of interesting observations which lack a clear message/flow. Some of the parts could be simply improved by a making an effort with the wording and the rationale of the experiments. But in many others, authors should make an effort strengthen the data.

We would like to thank for the reviewer’s positive comment about our experimental design. We have now rewritten our revised manuscript to improve the message flow and explain the rationale of the experiments with more details. Please see the highlighted areas throughout the maintext.

1. How do authors explain that high HK2 expression reduces vessel diameter? They claim that that HK2 regulates cell contractility via ROCK expression, but this is hardly convincing from the data provided.

We would like to thank for the reviewer’s comment. Please notice that previous studies indicate that RhoA-ROCK (Rho-associated protein kinase) signaling pathway has an important role in regulating retinal pericyte contractility (Durham et al., Am J Physiol Cell Physiol 2014; Kutcher et al., Am J Pathol 2007), while Kutcher et al., show that pericyte contractility determines endothelial cell cycle progression and sprouting. In addition, He et al., has reported that targeting α -SMA positive pericyte contractility could normalize tumor vasculature in a spontaneous pancreatic cancer mouse model (He et a., J Pathol 2018) (Please also see result and discussion section, page 10-11 line 249-259, page 23-24 line 614-645). In addition, Ito et al., 2004 show that ROCK inactivates the myosin light chain 2 phosphatase, which leads to increased phosphorylation of MLC2 and its mediated cell contractility, while ROCK can directly phosphorylate MLC2 too (Vicente-Manzanares et al., Nat. Rev. Mol. Cell Biol. 2009). In the previous version of manuscript, we have already shown that pharmaceutical inhibition of HK2 significantly rescues the defective blood vessel supporting function of TPC in vitro and in vivo, while it also down-regulates the expression of ROCK2 and p-MLC2 in TPC (Please see fig. 5 and 6, new supplementary fig. 6a-e). In addition, pre-treatment of HK2 or ROCK inhibitor significantly reduced carbachol-induced cell contraction in TPC as compared to placebo pre-treated TPC (Please see new supplementary fig. 6c and result page 10-11 line 249-259, and discussion page 24 line 637-639). To further demonstrate the role of HK2-ROCK2 mediated contractility in regulating blood vessel diameter, we now stably knock-downed HK2, ROCK1, ROCK2 expression in TPC and used these cells for in vitro tube formation assays and co-injection animal experiments. Our tube formation assay shows that depletion of pericyte-HK2 or -ROCK2 by shRNA/an inhibitor (i.e. a more potent ROCK2 inhibitor HA-1077; Dyberg et al., PNAS 2017) can enhance the total tube length, branch points and number in HUVEC co-cultured with TPC as compared with HUVEC co-cultured with scramble transfected TPC, while overexpression of ROCK2 in HK2-depleted TPC reduces the enhanced total tube length, branch points and number observed in HUVEC co-cultured with HK2-depleted TPC (Please see new supplementary fig. 6h-s). Importantly, depletion of pericyte-HK2 by shRNA also reduces

ROCK2 expression in TPC, but have no effect on ROCK1 expression (Please see new supplementary fig. 6h). Together, our results prove the importance of pericyte-HK2 expression/activity in regulating ROCK2 mediated cell contractility in vitro. Please also see result and discussion sections, page 14-15 line 366-384, page 22-24 line 580-645.

To confirm the role of pericyte-HK2 in modulating tumor blood vessel perfusion, diameter and subsequent drug delivery, we subcutaneously co-injected A549 (i.e. NSCLC cell line)/MHCC-LM9 (i.e. HCC cell line) cells with either HK2-depleted TPC, ROCK2 overexpressing HK2-depleted TPC or scramble transfected TPC into nude mice and then treated the tumor bearing mice with placebo or doxorubicin. Our data show that administration of doxorubicin reduces tumor growth in mice co-injected with A549/MHCC-LM9 cells and HK2-depleted TPC as compared to mice co-injected with A549/MHCC-LM9 cells and scramble transfected TPC (New supplementary figure 10a-h), while co-injection of A549/MHCC-LM9 cells and ROCK2 overexpressing HK2-depleted TPC decreases the enhanced doxorubicin efficacy against tumor growth observed in mice co-injected with A549/MHCC-LM9 cells and HK2-depleted TPC (New supplementary figure 10a-h). No significant difference in total blood vessel number and pericyte coverage is observed between groups (please see new supplementary fig. 10i-k). Interestingly, the tumor blood vessel diameter, perfusion and collagen IV staining, intratumoral Hoechst and doxorubicin level is all increased in the mice co-injected with A549 cells and HK2-depleted TPC as compared to the mice co-injected with A549 cells and scramble transfected TPC, while the increased blood vessel diameter, perfusion and collagen IV staining, intratumoral Hoechst and doxorubicin observed is reduced in mice co-injected with A549 cells and ROCK2 over-expressing HK2-depleted TPC (Please see new supplementary figure 10i-o, q). Furthermore, the expression of pericyte-p-MLC2 is reduced in tumors arising from the mice co-injected with A549 and HK2-depleted TPC as compared to A549 cells co-injected with scramble transfected TPC, whilst it is increased in the tumors derived from co-injection of A549 cells and ROCK2 overexpressing HK2-depleted TPC (New supplementary figure 10p). Further doppler ultrasound analysis of microbubble perfusion reveals that depletion of HK2 in TPC enhances A549 tumor blood flow and perfusion in mice as compared with scramble control group (please see new supplementary fig. 10r), while overexpression of ROCK2 in HK2-depleted TPC reduces the enhanced tumor blood flow and perfusion observed (Please see new supplementary fig. 10r). Overall, these new results further demonstrate the important role of pericyte-HK2 driven ROCK2-MLC2 mediated contractility in tumor vasculature remodelling and drug delivery in both lung and liver cancers. Please also see result section page 14-15 line 366-384, page 16-20, line 422-541 and discussion section page 25-26 line 663-707.

2. CD146 staining seems very weak, especially in NSLCC, at least in the pictures showed in Figure 1B. Only the depicted big vessel in the inset seems to be positive for CD146 (yellow), the other two are only red (α SMA positive) and the majority of the small vessels are only CD34 positive. Given that the MACS enrichment strategy is based on CD146 expression in vascular cells, this picture generates doubts about the percentage of pericytes obtained after FACS purification (Figure 1F and H). Also, this percentage is almost double in tumor samples compared with normal tissue. More images and a quantification of CD146 expression in tissue sections of normal and tumoral section will help to clarify this issue.

We apologize for the quality of our images. We now include more representative images of CD146/CD34/ α -SMA-triple immunostaining from HCC and NSCLC tissues and do a quantification of CD146 and α -SMA-double positive (+ve) blood vessels in the sections of normal adjacent tissues and tumors from our NSCLC and HCC patient cohorts respectively (n= 10 NSCLC or HCC patients), showing that around 65-72% of the blood vessels are CD146 and α -SMA-double positive in both normal adjacent tissues and tumors from NSCLC and HCC patients respectively, while no significant difference in the percentage of CD146 and α -SMA-double positive blood vessels between normal adjacent tissues and tumors is observed. Please see new figure 1a-f. For the FACS analysis, we now replace the pictures with better representative images of our FACS staging strategy in new figure 1h.

3. It seems that pericytes in vitro behave quite similarly irrespective of the origin. Are isolated pericytes losing their properties/dedifferentiating when cultured? However, when combined with endothelial cells, TPC seem to enhance tube/EC contraction compared to NPC. Again, the data are not convincing. How do authors explain that TPC express higher HK2 levels?

We would like to thank for the reviewer's interesting comment. In previous studies, most of the isolated pericytes are often not well-characterized (i.e. lack of pericyte characterization and functional test data) and maintained in either DMEM based media or medium not specified for pericyte growth condition (Stratman et al., Blood 2009; Orledge et al., J Cell Biol 1987). In contrast, we have cultured and maintained our isolated pericytes derived from lung and liver tissues, using commercially available human pericyte medium (ScienCell Research Laboratories-#1201). We have also performed a comprehensive characterization experiment confirming the pericyte identity and function of our isolated NPC and TPC (Please see fig 1, 2, 3, supplementary fig. 2, 3, 4). We now also show that there is no significant difference in the mural cell/pericyte marker expression between TPC at 3rd and 12 passages (Please see new supplementary fig. 3d, e), indicating that our culturing method can maintain the pericyte identity of our isolated NPC and TPC during passages. Please also see new result page 8 line 187-190, and discussion page 21-22 line 555-578.

Although Hosaka et al indicate that pericyte undergoes fibroblast transition after the stimulation of PDGF-BB (Hosaka et al, PNAS 2016), a recent study shows that pericytes of multiple organs do not behave as mesenchymal stem cells in vivo (Guimaraes-Camboa et al., Cell Stem Cell 2017). Since the MSC properties/dedifferentiating ability of pericytes remains debating in the field, we believe that it is currently out of scope for us to examine this argument in this manuscript. Instead, we focus on developing a new isolation method of NPC and TPC derived from NSCLC and HCC tissues and examining their blood vessel supporting role only. Please see new discussion section page 21-22 line 555-578.

For the tube formation assay, we now provide new data showing that stable knock-down of pericyte-HK2 or depleting pericyte-ROCK2 by shRNA/ROCK inhibitor HA-1077 (i.e. a more potent ROCK2 inhibitor; Dyberg et al., PNAS 2017) increases the total tube length, branch points and number in HUVEC co-cultured with TPC as compared with HUVEC co-cultured with scramble transfected TPC (Please see new supplementary fig. 6h-s), while overexpression of ROCK2 in HK2-depleted TPC significantly reduces the enhanced total tube length, branch points and number observed in HUVEC cocultured with HK2-depleted TPC (Please see new supplementary fig. 6h-s). Importantly, depletion of pericyte-HK2 expression by shRNA also reduces ROCK2 expression in NSCLC/HCC derived TPC (Please see new supplementary figure 6h), which is same as the effect of HK2 inhibitor 3-BP on TPC as we already reported (Figure 5a-j). Overall, we provide strong evidence that HK2 expression regulates the blood vessel supporting function of TPC via ROCK2 mediated contractility. Please also see result page 14-15 line 366-384, and discussion page 24 line 639-645.

For the HK2 up-regulation's question, we have already shown that the conditioned medium from A549/HepG2 cells can up-regulate HK2 and ROCK2 expression as well as glycolysis in normal tissue derived pericytes (NPC), which causes them becoming defective in their blood vessel supporting role (Fig. 5p-t, new supplementary fig. 6t, u), suggesting that tumor cells regulate pericyte's blood vessel supporting function via paracrine signals. Coincidentally, previous studies indicate that tumor cells can regulate stromal cell metabolism (Schworer et al., Cell Metabolism 2019), while Chen et al show that tumor cells derived hyaluronan (HA) fragments can activate glycolytic gene expression, including HK2, in monocytes (Journal of Hepatology. 2019, 71, 333-343). Interestingly, we provide new data showing that NSCLC/HCC derived NPC express the HA receptor CD44 on their cell surface. Furthermore, we now show that blockade of HA by a HA antagonist (Pep-1; Chen et al., Journal of Hepatology. 2019, 71, 333-343) prohibits the up-regulation of HK2 and ROCK2 expression in NPC after treated with conditioned medium from A549/HepG2 cells, suggesting that tumor-derived HA fragments enhance pericyte-HK2 driven glycolysis to dysregulate pericyte's blood vessel supporting role

by activating ROCK2 mediated contractility. Please also see new supplementary fig. 6v, w, result and discussion section, page 15-16 line 398-411 and page 24-25 line 652-661.

4. Figure 4J. Authors claim that HK2 expression is significantly increased in mural cells in tumors. Quantification of that expression in representative images should be provided. Endothelial cells seem to be contributing to HK2 overexpression too, as well as many other tumoral/stromal cells. This observation compromises the pericyte-driven hypothesis of the effects of in vivo, pharmacological inhibition of HK2. Pre-treated or genetically modified pericytes should be implanted together with tumor cells or, ideally, tumor cells should be implanted in pericyte-specific HK2 deficient mice.

We would like to thank for the reviewer's suggestion. We have now provided the quantification of mural-HK2 expression in normal adjacent tissue and tumors derived from NSCLC and HCC patients (n= 10 NSCLC/HCC patients). Please see new fig. 4j and supplementary fig. 5d.

To answer the question about the observed synergistic effect of HK2 inhibitor and doxorubicin in vivo is due to direct inhibition on tumor cells or through modulating TPC function, as suggested by reviewer 2, we now provide new data showing that co-administration of 3-BP and doxorubicin has no synergistic inhibitory effect on LLC tumor cell growth in vitro (Please see new supplementary fig. 8t). In addition, exposure of NPC with conditioned medium derived from 3-BP or/and doxorubicin pre-treated cancer cells can still up-regulate pericyte-HK2 and -ROCK2 expression as compared with untreated NPC (Please see new supplementary fig. 9). These new data suggest that the synergistic inhibitory effect of 3-BP and chemotherapeutic agent on tumor growth is through modulating TPC's blood vessel supporting function, but not via direct inhibition on tumor cells. Please also see result and discussion page 18 line 477-493, page 25 line 675-684.

To further demonstrate the role of pericyte-HK2 in modulating tumor vasculature and subsequent drug delivery, as suggested by the reviewer 1, we subcutaneously co-injected A549(i.e. NSCLC cell line)/MHCC-LM9(i.e. HCC cell line) cells with either HK2-depleted TPC, ROCK2 overexpressing HK2-depleted TPC or scramble transfected TPC into nude mice and then treated the tumor bearing mice with placebo or doxorubicin. Strikingly, administration of doxorubicin reduces tumor growth in mice co-injected with A549/MHCC-LM9 cells and HK2-depleted TPC as compared to the mice co-injected with A549/MHCC-LM9 cells and scramble transfected TPC (Please see new supplementary fig. 10a-h), while co-injection of cancer cells with ROCK2 overexpressing HK2-depleted TPC reduces the enhanced doxorubicin efficacy against tumor growth observed in mice co-injected with A549/MHCC-LM9 cells and HK2-depleted TPC (Please see new supplementary fig. 10a-h). No significance difference in total blood vessel number and pericyte coverage was observed between groups (Please see new supplementary fig. 10i-k). Interestingly, the tumor blood vessel diameter and perfusion, collagen IV staining and intratumoral Hoechst is increased in mice that are co-injected with A549 cells and HK2-depleted TPC as compared to mice co-injected with A549 cells and scramble transfected TPC, while the increased blood vessel perfusion/functions observed is reduced in mice co-injected with ROCK2 overexpressing HK2-depleted TPC and A549 cells (please see new supplementary figure 10l-o). Furthermore, the expression of pericyte-p-MLC2 is reduced in tumors arising from the mice co-injected with A549 and HK2-depleted TPC as compared to A549 co-injected with scramble transfected TPC, whilst its expression is increased in the tumors derived from co-injection of ROCK2 overexpressing HK2-depleted TPC and A549 cells (please see new supplementary figure 10p). Importantly, ex vivo two photon imaging shows that the tumor blood vessel diameter and doxorubicin delivery are increased in the mice co-injected with A549 and HK2-depleted TPC as compared with the mice co-injected with A549 and scramble transfected TPC, while the increased blood vessel diameter and doxorubicin delivery observed is decreased in mice co-injected with ROCK2 overexpressing HK2-depleted TPC and A549 cells (Please see new supplementary figure 10q). Doppler ultrasound analysis of microbubble perfusion reveals that depletion of HK2 in TPC increases A549 tumor blood flow and perfusion in mice as compared with control group (please see new supplementary fig. 10r), while overexpression of ROCK2 in HK2-depleted TPC reduces the

enhanced tumor blood flow and perfusion observed (Please see new supplementary fig. 10r). These results indicate the important role of pericyte-HK2 driven ROCK2-MLC2 mediated contractility in tumor vasculature remodelling and drug delivery in lung and liver cancers. Please also see result section page 19-20 line 495-541, and discussion section page 25-26 line 684-699.

5. Conclusions obtained from clinical correlation data (Figure 4K, Suppl Fig 4E) on HK2 expression are not valid, markers used are not specific of pericytes. PDGFR β and α SMA are expressed by cancer fibroblasts and CD146 is also expressed in endothelial cells.

We would like to thank for the reviewer's comment. Please notice that these markers are widely used to study the effect of mural cells (including pericytes) on psychological and pathological studies (Wong et al., Cell 2020; Lectuerier et al., Nature Communications 2020; Dieguez-Hurtado et al., Nature Communications 2019; Chen et al., Protein and Cell 2018). To solidify our finding, we now perform triple-immunostaining of HK2, α -SMA and CD34 with tumor sections derived from our NSCLC and HCC patient cohorts (n= 77 NSCLC patients; n= 101 HCC patients with complete clinical data) instead, confirming the correlation between mural-HK2 expression and patient prognosis in NSCLC and HCC patients respectively. Please see new figure 4k and supplementary figure 5e, result and discussion page 12, line 299-310 and page 23, line 610-614.

6. Data on PDGFR β expression in cultured cells in Figure 2B does not match the qPCR profiles, as fibroblasts in the IF image show much greater expression than qPCR graph depicts compared with pericytes

We would like to thank for the reviewer's suggestion. We have now replaced the image with a better representative picture of PDGFRB staining in fibroblasts. Please see new fig. 2b.

Reviewer #2 (Remarks to the Author); expert on metabolism:

Pericytes play an important role in angiogenesis and tumor microenvironment formation. The authors developed a novel method to isolate pericytes from human normal and tumor tissues. This is a robust model for studying the role of pericytes in angiogenesis and cancer progression. The authors discovered that pericytes from tumor had higher glycolytic flux but lower mitochondrial activity, suggesting that the TPC also display the Warburg effect, which alters cell contractility and limit blood supply. This could further exacerbate tumor hypoxia.

We would like to thank for the reviewer's positive feedback about the novelty and robustness of our newly developed pericyte model.

A few additional experiments that could strengthen the conclusions:

In 125-128, how did the authors conclude that 'CD146 was expressed in both endothelial cells and pericytes ..., while CD34 and PDGFR β was specifically expressed in endothelial cells and pericytes respectively'? Were independent markers used to confirm those cells are endothelial cells and pericytes respectively? How can they tell that CD146 was expressed in pericyte without showing the co-staining of CD146 and PDGFR β in Figure1?

We are sorry for not explaining the selection of endothelial/pericyte markers used in this study clearly. We have now rewritten the result section to make it more clear for the logic and rational behind our marker selection. Please see result page 6-7 line 125-146 and below. For your interest, many studies have already shown that CD146 is expressed in both endothelial cells and pericytes (Chen et al., Protein and Cell 2018; Chen et al., PNAS 2017). "Since anti-human CD146 antibody conjugated with FITC and anti-FITC magnetic microbeads are commercially available, we proposed to employ a CD146+-FITC-microbead activated cell sorting (MACS) method to enrich the vascular cell population from tissue samples first, and then used positive and negative selection markers to isolate pericyte population via fluorescence activated cell sorting (FACS). In order to carry out our pericyte purification method, we first examined whether CD146 was expressed on vascular endothelial cells and pericytes within normal

adjacent tissues and tumors derived from NSCLC and HCC patients. Due to the fact that CD34 and α -SMA/PDGFR β is known as a vascular endothelial cell and mural cell/pericyte marker respectively (Wong et al., Cell 2020; Lectuerier et al., Nature Communications 2020; Dieguez-Hurtado et al., Nature Communications 2019; Siemerink et al., Angiogenesis 2012; Xu et al., Clinical Cancer Research 2017; Clare M. Isacke, in The Adhesion Molecule FactsBook (Second Edition), 2000), we therefore used them to define the CD146 staining in this study. We first performed triple immunofluorescent staining of these markers with paired normal adjacent tissues and tumors derived from NSCLC and HCC patients respectively, showing that CD146 is expressed in both CD34 positive (+ve) vascular endothelial cells and α -SMA/PDGFR β positive pericytes within normal adjacent tissues and tumor tissues derived from NSCLC and HCC patients, while around 65-72% percentage of blood vessels are CD146 and α -SMA double positive in both normal adjacent tissues and tumors derived from NSCLC and HCC patients respectively (new Fig. 1a-f, new supplementary fig. 2a, b). No significant difference in the percentage of CD146 and α -SMA-double positive blood vessels is observed between normal adjacent tissues and tumors (Fig. 1c, f). Therefore, our results suggest that CD146+-FITC microbead activated cell sorting method could be used to enrich the vascular cell population from tissue samples derived from NSCLC and HCC patients". Furthermore, we have already performed a comprehensive characterization study, showing that our isolated NPC and TPC express good level of pericyte markers such as PDGFR β , CD146, CD13, Desmin, but express relatively low or undetectable levels of endothelial cell markers, immune cell maker, fibroblast marker and smooth muscle cell markers (Please see fig 2a-c, new supplementary fig. 3a-e). Based on the reviewer's suggestion, we now also provide new triple immunostaining data with our tissue sections, indicating that CD146 is co-expressed with PDGFR β in pericytes within normal adjacent tissues and tumors from NSCLC and HCC patients. Please see new supplementary fig. 2a, b and introduction page 3 line 56-63, result page 6-7 line 125-146.

The data demonstrated a strong association between HK2/glycolysis upregulation and ROCK2 activation. However, it is not clear whether the paracrine factors from tumor cells reprogram metabolism to activate ROCK2, or activate ROCK2 to reprogram metabolism. Does ROCK2 inhibitor represses the Warburg effect in TPCs?

We would like to thank for the reviewer's comment. Previous studies indicate that tumor cells can regulate stromal cell metabolism (Schworer et al., Cell Metabolism 2019), while Chen et al show that tumor cells derived hyaluronan (HA) fragments can activate glycolysis in monocytes, which can be inhibited by using a HA antagonist (Pep-1) (Chen et al., Journal of Hepatology. 2019, 71, 333-343). Coincidentally, we provide new data showing that NSCLC/HCC derived NPC express the HA receptor CD44 on their cell surface, while blockade of HA by a HA antagonist (Pep-1) prohibits the up-regulation of HK2 and ROCK2 expression in NPC after exposed with conditioned medium from A549/HepG2 cells (Please see new supplementary fig. 6v, w), suggesting that tumor cells regulate pericyte-HK2 and ROCK2 expression probably via the HA-CD44 signalling. Notably, no significant effect on the expression of ROCK1 in NPC after exposed with conditioned medium from A549/HepG2 cells in the presence or absence of control peptide or HA antagonist was observed (Please see new supplementary fig. 6h, v, w). To determine whether ROCK inhibitor represses the Warburg effect in TPCs, we now perform lactate concentration measurement and seahorse assays as well as RT-PCR and western blot analysis, showing that treatment with ROCK inhibitor does not affect HK2 expression and glycolysis (i.e. Warburg effect) in NSCLC/HCC derived TPC (please see new supplementary fig. 7a-g). For your interest, we now also show that depletion of HK2 by shRNA down-regulates ROCK2 expression in NSCLC/HCC derived TPC (New supplementary figure 6h). Altogether, these findings suggest that tumor-derived paracrine signal activates HK2 driven pericyte metabolic reprogramming to promote ROCK2-ML2 mediated cell contractility. In addition, ROCK2 is the downstream effector of HK2-mediated glycolysis in TPC. Please also see new result and discussion sections page 14 line 368-373, page15-16 line 398-420 and page 24-25 line 652-661.

3BP and Dox had synergistic effect in repressing tumor growth in vivo. It is difficult to dissect whether it was due to direct inhibition of tumor cells, or through modulating TPC functions. Does 3BP and Dox combination inhibit LLC cell growth in vitro? Can the CM from 3BP and Dox treated tumor cells activate HK2 and ROCK2 in pericytes?

We would like to thank for the reviewer's comment. We now provide new data indicating that 3BP and Dox combination shows no synergistic inhibitory effect on LLC cell growth in vitro (please see new supplementary fig. 8t), while exposure of NSCLC/HCC derived NPC with conditioned medium from 3BP or/and DOX pre-treated tumor cells can still up-regulate HK2 and ROCK2 expression in NPC as compared to untreated NPC (please see supplementary fig. 9). These data suggest that the synergistic inhibitory effect of 3-BP and Dox on tumor growth observed in vivo is due to the modulation of TPC function. In the revised manuscript, we also performed co-injection of cancer cells with either HK2-depleted TPC, ROCK2 overexpressing HK2 depleted TPC or scramble transfected TPC into nude mice and treated the tumor bearing mice with placebo or doxorubicin, indicating that depletion of HK2 in TPC significantly enhances blood vessel perfusion and diameter, thereby increasing doxorubicin delivery and efficacy against tumor growth, while overexpression of ROCK2 in HK2-depleted TPC reduces the enhanced vascular effect and doxorubicin delivery/efficacy against tumor growth observed, indicating that pericyte-HK2 expression regulates its blood vessel supporting function and subsequent drug delivery via ROCK2 (Please see new supplementary figure 10). In addition, our new tube formation assay also shows that depletion of HK2 in TPC increases HUVEC tube length, branch points and number as compared to HUVEC cocultured with scramble transfected TPC, while overexpression of ROCK2 in HK2 depleted TPC reduces the enhanced effect observed (Please see new supplementary fig. 6h-s). Overall, these new data demonstrate that depletion of pericyte-HK2 enhances blood vessel function and subsequent drug delivery to prohibit cancer growth. Please also see result and discussion section page 14-15 line 366-384, page 18-20 line 477-541 and page 25-26 line 663-707.

Reviewer #3 (Remarks to the Author); expert on ROCK2:

This paper nicely demonstrates the distinct differences between normal and tumour associated pericyte function in angiogenesis and demonstrates the use of new isolation protocol to facilitate a deeper understanding of the differences between their signalling in vitro and recapitulated in vivo. Moreover, they reveal distinct differences in glycolysis and the role the ROCK2-MLC2 signalling plays in vasculature normalisation and importantly capitalise upon this to show that manipulation of this can improve chemotherapy delivery.

We would like to thank for the reviewer's positive comment about our findings.

Can the authors perform some shRNA experiments of manipulation to show ROCK 2 effect are key to this work rather than ROCK1 off target effect- this should be easy enough to perform to demonstrate that ROCK2 rather than pan drugs such as GSK429286A are having a role in improving chemotherapy delivery beyond off target ROCK1 vascular relaxation effects?

Any other rock2 specific manipulation to address the above point would also satisfy this issue.

We would like to thank for the reviewer's suggestion. We now provide new tube formation assay data showing that depletion of ROCK2 by shRNA or a ROCK inhibitor HA-1077 (i.e. a more potent ROCK2 inhibitor; Dyberg et al., PNAS 2017) increases the total tube length, branch points and number in HUVEC co-cultured with TPC as compared with HUVEC co-cultured with scramble transfected TPC (Please see new supplementary fig. 6h-u), while overexpression of ROCK2 in HK2-depleted TPC significantly reduces the enhanced total tube length, branch points and number observed in HUVEC cocultured with HK2-depleted TPC (Please see new supplementary fig. 6h-s). In contrast, depletion of ROCK1 by shRNA in TPC did not affect HUVEC tube length, branch points and number as compared to HUVEC cocultured with scramble transfected TPC. Importantly, depletion of pericyte-HK2 expression by shRNA reduces ROCK2 expression in NSCLC/HCC derived TPC, while it has no effect on ROCK1

expression (Please see new supplementary figure 6h). In vivo, we now show that administration of doxorubicin reduces tumor growth in mice co-injected with HK2-depleted TPC and A549 (i.e. NSCLC cell line)/MHCC-LM9 (i.e. HCC cell line) cells as compared to mice co-injected with scramble transfected TPC and A549/MHCC-LM9 cells, while co-injection of ROCK2 overexpressing HK2-depleted pericytes with A549/MHCC-LM9 cells reduces the enhanced doxorubicin efficacy against tumor growth observed in mice co-injected with HK2-depleted pericytes and A549/MHCC-LM9 cells (Please see new supplementary fig. 10 a-h). In addition, the tumor blood vessel diameter, perfusion and collagen IV staining, intratumoral Hoechst and doxorubicin level is all increased in mice co-injected with A549 cells and HK2-depleted TPC as compared to mice co-injected with A549 cells and scramble transfected TPC, while the increased blood vessel perfusion, diameter, collagen IV staining, intratumoral Hoechst and doxorubicin level observed is reduced in mice co-injected with A549 cells and ROCK2 overexpressing HK2-depleted TPC cells (please see new supplementary fig. 10i-o, q-r). In addition, the expression of pericyte p-MLC2 is reduced in tumors arising from mice co-injected with A549 cells and HK2-depleted TPC as compared to mice co-injected with A549 cells and scrambled transfected TPC, while it is increased in the tumors derived from mice co-injected with A549 cells and ROCK2 overexpressing HK2-depleted TPC (please see new supplementary figure 10p) . Overall, our result demonstrates that ROCK2, but not ROCK1, is the downstream effector of pericyte-HK2 mediated blood vessel supporting function. Please also see result and discussion sections page 14-15 line 366-384, page 19-20 line 495-541, page 24 line 639-645, and page 25 line 684-699.

Lastly can the authors perform experiments that the effect they see in vivo for LLC (ie improve chemotherapy response) also hold true for HCC cancer as this is a major (ie half of the paper) aspect of this work which is not show as a final in vivo experiment?

We would like to thank for the reviewer's suggestion. We now show that treatment with 3BP and doxorubicin significantly inhibits the growth of mouse HCC cell line Hep-1-6 as compared to placebo, 3BP or doxorubicin alone, confirming the importance of our finding in both lung and liver cancer. Please see new supplementary fig. 8e-h, result and discussion sections, page 17 and 25. For your interest, we also provide new data indicating that administration of doxorubicin reduces tumor growth in mice subcutaneously co-injected with HK2-depleted TPC and MHCC-LM9 cells (i.e. human HCC cell line) as compared to the mice subcutaneously co-injected with scramble transfected TPC and MHCC-LM9 cells (Please see new supplementary fig. 10a-h), while co-injection of ROCK2 overexpressing HK2-depleted pericytes with MHCC-LM9 cells reduces the enhanced doxorubicin efficacy against tumor growth observed in mice co-injected with HK2-depleted pericytes and MHCC-LM9 cells (Please see new supplementary fig. 10a-h). These results indicate the important role of pericyte-HK2 driven ROCK2-MLC2 mediated contractility in tumor vasculature remodelling and drug delivery and efficacy against liver cancer growth too. Please also see result section page 17-18 line 422-437, page 19 line 495-507 and discussion section page 25-26 line 667-672, line 684-692.

REVIEWER COMMENTS

Reviewer #1 (Remarks to the Author):

Overall, this manuscript has greatly improved, specially regarding Figures 6. However, I still have some minor concerns.

Authors still do not answer my question on how HK2 regulates Rock expression/contractility. They only provide nice associations. This should at least be discussed.

aSMA and CD146 do not seem to co-localize. Instead, CD146 and Pdgfrb do (both in normal tissue and tumour). Is there an explanation for that?

The red colour in the graphs in Figure 5 (TPC + 3-BP) should be changed, as it is misleading with the NPC in Figure 3.

It would be nice to prove that pMLC is high in TPC in vivo, like in Fig. 4J for HK2

These 2 papers should be acknowledged (cited) in this study, as they are relevant in the context of the data (pericytes and angiogenesis).

<https://pubmed.ncbi.nlm.nih.gov/29307447/>
<https://pubmed.ncbi.nlm.nih.gov/32466671/>

In general, the text should be edited as it often contains very long sentences that are difficult to follow. See as an example (page 6, lines 131-139 or page 12, lines 303-310).

Reviewer #2 (Remarks to the Author):

The authors have done a great work addressing the previous critiques. It looks ready for publication as far as I can tell.

Please label when and what mitochondrial uncoupler and inhibitors were added in figure 4G and H. And in Figure S7 B,C,D,E.

Reviewer #3 (Remarks to the Author):

The revised MS addresses all my requests and is now ready for publication.

Reviewer #1 (Remarks to the Author):

Overall, this manuscript has greatly improved, specially regarding Figures 6.

However, I still have some minor concerns.

We would like to thank for the reviewer's positive support about our findings and manuscript.

Authors still do not answer my question on how HK2 regulates Rock expression/contractility.

They only provide nice associations. This should at least be discussed.

We would like to thank for the reviewer's comment. Please notice that RhoA-ROCK (Rho-associated protein kinase) signaling pathway has been shown to play an important role in regulating retinal pericyte contractility (Durham et al., Am J Physiol Cell Physiol 2014; Kutcher et al., Am J Pathol 2007), while Kutcher et al., show that pericyte contractility determines endothelial cell cycle progression and sprouting. In addition, He et al., has reported that targeting α -SMA positive pericyte contractility could normalize tumor vasculature in a spontaneous pancreatic cancer mouse model (He et al., J Pathol 2018) (Please also see result and discussion section, page 13-14 line 340-347, page 23-24 line 627-633). In addition, Ito et al., 2004 show that ROCK inactivates the myosin light chain 2 phosphatase, which leads to increased phosphorylation of MLC2 and its mediated cell contractility, while ROCK can directly phosphorylate MLC2 too (Vicente-Manzanares et al., Nat. Rev. Mol. Cell Biol. 2009). In our manuscript, we have shown that pharmaceutical inhibition of HK2 significantly rescues the defective blood vessel supporting function of TPC in vitro and in vivo, while it also down-regulates the expression of ROCK2 and p-MLC2 in TPC (Please see fig. 5 and 6, supplementary fig. 6). In addition, pre-treatment of HK2 or ROCK inhibitor reduce carbachol-induced cell contraction in TPC as compared to placebo pre-treated TPC (Please see supplementary fig. 6c and result page 13 line 334-336, and discussion page 24 line 636-638). Our tube formation assay shows that depletion of pericyte-HK2 or -ROCK2 by shRNA/an inhibitor (i.e. a more potent ROCK2 inhibitor HA-1077; Dyberg et al., PNAS 2017) can enhance the total tube length, branch points and number in HUVEC co-cultured with TPC as compared with HUVEC co-cultured with scramble transfected TPC, while overexpression of ROCK2 in HK2-depleted TPC reduces the enhanced total tube length, branch points and number observed in HUVEC co-cultured with HK2-depleted TPC (Please see supplementary fig. 6h-s). Importantly, depletion of pericyte-HK2 by shRNA also reduces ROCK2 expression in TPC, but have no effect on ROCK1 expression (Please see supplementary fig. 6h). Together, our results prove the importance of pericyte-HK2 expression/activity in regulating ROCK2 mediated cell contractility in vitro. Please also see result and discussion sections, page 14-15 line 365-383, page 23-24 line 613-644.

To confirm the role of pericyte-HK2 in modulating tumor blood vessel perfusion, diameter and subsequent drug delivery, we subcutaneously co-injected A549 (i.e. NSCLC cell line)/MHCC-LM9 (i.e. HCC cell line) cells with either HK2-depleted TPC, ROCK2 overexpressing HK2-depleted TPC or scramble transfected TPC into nude mice and then treated the tumor bearing mice with placebo or doxorubicin. Our data show that administration of doxorubicin reduces tumor growth in mice co-injected with A549/MHCC-LM9 cells and HK2-depleted TPC as compared to mice co-injected with A549/MHCC-LM9 cells and scramble transfected TPC (Supplementary figure 10a-h), while co-injection of A549/MHCC-LM9 cells and ROCK2 overexpressing HK2-depleted TPC decreases the enhanced doxorubicin efficacy against tumor growth observed in mice co-injected with A549/MHCC-LM9 cells and HK2-depleted TPC (Supplementary figure 10a-h). No significant difference in total blood vessel number and pericyte coverage is observed between groups (please see supplementary fig. 10i-k). Interestingly, the tumor blood vessel diameter, perfusion and collagen IV staining, intratumoral Hoechst and doxorubicin level is all increased in the mice co-injected with A549 cells and HK2-

depleted TPC as compared to the mice co-injected with A549 cells and scramble transfected TPC, while the increased blood vessel diameter, perfusion and collagen IV staining, intratumoral Hoechst and doxorubicin observed is reduced in mice co-injected with A549 cells and ROCK2 over-expressing HK2-depleted TPC (Please see supplementary figure 10i-o, q). Furthermore, the expression of pericyte-p-MLC2 is reduced in tumors arising from the mice co-injected with A549 and HK2-depleted TPC as compared to A549 cells co-injected with scramble transfected TPC, whilst it is increased in the tumors derived from co-injection of A549 cells and ROCK2 overexpressing HK2-depleted TPC (Please see supplementary figure 10p). Further doppler ultrasound analysis of microbubble perfusion reveals that depletion of HK2 in TPC enhances A549 tumor blood flow and perfusion in mice as compared with scramble control group (please see supplementary fig. 10r), while overexpression of ROCK2 in HK2-depleted TPC reduces the enhanced tumor blood flow and perfusion observed (Please see supplementary fig. 10r). Overall, these results further demonstrate the important role of pericyte-HK2 driven ROCK2-MLC2 mediated contractility in tumor vasculature remodelling and drug delivery in both lung and liver cancers. Please also see result section page 16-20, line 421-540 and discussion section page 25-26 line 683-698.

For the reviewer's interest, recent studies indicate that glycolysis can potentially regulate gene expression via histone modification (Wu et al., Journal of Genetics and Genomics, 2019 Dec 20;46(12):561-574), while its potential role in our case can be examined in the future study. Since we have already provided solid evidence proving the association between glycolysis and ROCK2 expression/contractility in our manuscript, we believe that it is out of the scope for us to examine this further in the current study.

aSMA and CD146 do not seem to co-localize. Instead, CD146 and Pdgfrb do (both in normal tissue and tumour). Is there an explanation for that?

We would like to thank for the reviewer's comment. Previous studies show that α -SMA is a mural cell marker (including smooth muscle cell, pericytes and fibroblasts (especially highly expressed in cancer associated fibroblast) (Armulik et al., Developmental Cell 2011; Wong et al., Cell 2020; Lechertier et al., Nature Communications 2020; Wong et al., Cancer Cell 2015), while CD146 is highly expressed in endothelial cells and pericytes and PDGFRB mostly expressed in vascular pericytes (Wong et al., Cell 2020; Chen et al., PNAS 2017; Leve' en, et al., Genes Development 1994). Consistent with these findings, our result show that the colocalization of α -SMA and CD146 are mostly observed around CD34+ve blood vessels in normal and tumor tissues, while there are also some α -SMA+ve/CD146-ve fibroblasts present in these tissues (Please see figure 1a-f). Since CD146 and PDGFRB are known as endothelial cell and/or pericyte markers (Wong et al., Cell 2020; Chen et al., PNAS 2017; Leve' en, et al., Genes Development 1994), the expression of CD146 and PDGFRB are therefore highly colocalized around CD34+ blood vessels within normal and tumor tissues (Please see supplementary figure 2a, b).

The red colour in the graphs in Figure 5 (TPC + 3-BP) should be changed, as it is misleading with the NPC in Figure 3.

We would like to thank for the reviewer's suggestion. We have now changed the color in the graphs in figure 5 (TPC+3BP). Please see new figure 5.

It would be nice to prove that pMLC is high in TPC in vivo, like in Fig. 4J for HK2

We would like to thank for the reviewer's suggestion. Please notice that we have already shown that the expression of p-MLC2 is up-regulated in NSCLC/HCC derived TPC in vitro (Figure 5j). In vivo, we also show that TPC expresses p-MLC2 in A549 and LLC subcutaneous tumors, while treatment with 3-BP can reduce pericyte-p-MLC2 expression

in subcutaneous LLC and A549 tumors. In addition, depletion of HK2 by shRNA also down-regulated pericyte-p-MLC2 expression in A549 tumors co-injected with TPC as compared with A549 tumors co-injected with scramble transfected TPC (Supplementary figure 8q and 10p). Overall, these data have already proven the role of p-MLC2 expression in regulating pericyte's function. As suggested by the reviewer, we have provided new data indicating that the percentage of mural-p-MLC2 positive blood vessels is significantly up-regulated in tumors as compared to normal adjacent tissues in both NSCLC and HCC patients. Please see the data below.

Figure: Representative images of the triple immunostaining of CD34 (magenta), p-MLC2 (green) and α -SMA (red) in normal adjacent tissues and tumors derived from NSCLC and HCC patients respectively. Bar charts represent the percentage of p-MLC2/ α -SMA double positive blood vessels per field in each group (n= 10 NSCLC or HCC patients).

These 2 papers should be acknowledged (cited) in this study, as they are relevant in the context of the data (pericytes and angiogenesis).

<https://pubmed.ncbi.nlm.nih.gov/29307447/>
<https://pubmed.ncbi.nlm.nih.gov/32466671/>

We would like to thank for the reviewer's suggestion. We have now cited these two publications in our main text. Please see page 3, line 52 and 66

In general, the text should be edited as it often contains very long sentences that are difficult to follow. See as an example (page 6, lines 131-139 or page 12, lines 303-310).

We would like to thank for the reviewer's comment. We have now restructured some of our sentences in page 7, lines 153-156 and page 13, lines 322-325.

Reviewer #2 (Remarks to the Author):

The authors have done a great work addressing the previous critiques. It looks ready for publication as far as I can tell.

We would like to thank for the reviewer's positive support for our manuscript to be published in Nature Communications.

Please label when and what mitochondrial uncoupler and inhibitors were added in figure 4G and H. And in Figure S7 B,C,D,E.

We would like to thank for the reviewer's comment. We have now labelled the mitochondrial uncoupler and inhibitors being used at specific time periods during the seahorse experiments in figure 4G/H and figure S7 B, C, D, E.

Reviewer #3 (Remarks to the Author):

The revised MS addresses all my requests and is now ready for publication.

We would like to thank for the reviewer's positive support for our manuscript to be published in Nature Communications.